# Generalizable Multilevel Graph Optimization via Reinforcement-Guided Diffusion from Simple Subproblems

## Abstract

Solving multilevel graph combinatorial problems (GCPs) is challenging due to their structural complexity and the limited generalization capabilities of existing learning-based optimization algorithms. Models trained on GCPs often fail to generalize to more complex or larger multilevel problems. We propose a Reinforcement Learning-Guided Diffusion Model (RLG-DM) that addresses this challenge by combining structural priors learned from simple problems. In the forward diffusion process, noise is progressively injected into the structure of a graph neural network, each representing and having been trained on a simple combinatorial optimization problem, such as the facility location or vehicle routing problem. In the reverse diffusion phase, a reinforcement learning controller guides the step-wise generation of subgraphs from a randomly initialized graph by dynamically selecting and combining the pretrained diffusion models based on task-specific hierarchies. Trained only on representative subproblems, the controller generalizes to unseen multilevel GCP tasks without retraining. We evaluate the proposed RLG-DM on representative multilevel GCPs, such as the location routing problem, the nurse rostering problem and flexible job shop scheduling. Experimental results show that RLG-DM consistently outperforms state-of-the-art baselines and generalizes effectively to structurally diverse, unseen tasks.

## 1 Introduction

Graph combinatorial problems (GCPs), such as location routing problems (LRPs) Yan et al. (2019), flexible job shop scheduling (FJSP) Ghaedy-Heidary et al. (2024); Smit et al. (2025), and nurse scheduling problems (NSPs) Valouxis et al. (2012), are central to many real-world applications in logistics, manufacturing, and workforce planning. These problems are typically NP-hard, involving discrete decision spaces, intricate dependencies, and exponential solution complexity Jin et al. (2025). They are often regarded as multilevel combinatorial problems because of their hierarchical structure and interdependent decision layers. Traditional approaches are mainly divided into exact and approximate methods Blum & Roli (2003): while exact solvers provide optimality guarantees, they become computationally infeasible as problem size increases; heuristic and metaheuristic methods (e.g., evolutionary algorithms Huang et al. (2023b), local search Bai et al. (2010)) scale better but rely heavily on handcrafted rules and lack generalization.

Recent advances in graph neural networks (GNNs) have shown strong potential in solving combinatorial problems by learning structural patterns from data Schuetz et al. (2022); Zhao et al. (2023). These models typically follow a two-stage pipeline: encoding the problem graph via message-passing mechanisms Cappart et al. (2023), followed by solution generation through learned decoders or policies Joshi et al. (2019); Khalil et al. (2017). GNNs offer a flexible, end-to-end alternative to handcrafted heuristics and can generalize across similar instances of flat GCPs, such as the traveling salesman problem (TSP) or vehicle routing problem (VRP). However, applying GNNs to multilevel GCPs remains a significant challenge. Problems such as location routing and nurse rostering often involve hierarchical decision structures and interacting subgraphs, which require not only learning local graph patterns but also managing global interdependencies He et al. (2025). Existing GNNs, when trained on flat or single-level tasks, struggle to coordinate decisions across multiple levels, limiting their scalability and effectiveness on multilevel GCPs. This limitation has motivated the

search for more expressive generative frameworks that can better capture the structured complexity of multilevel problems.

Diffusion models (DMs) have recently emerged as a promising class of generative models for structured optimization Yan & Jin (2024); Li et al. (2025); Sun & Yang (2023). By learning to reverse a noise-injection process, DMs can progressively generate high-quality solutions that preserve problem-specific structural patterns. When applied to GCPs, DM-based approaches such as DI-FUSCO Sun & Yang (2023) and its extensions Li et al. (2024); Huang et al. (2023a) have demonstrated notable success in producing feasible solutions in an end-to-end manner. Crucially, DMs offer a complementary strength to GNNs, as their generative nature enables them to capture global structures that GNNs alone may overlook, especially in complex multilevel settings. However, despite their expressive modeling capacity, most existing DMs are trained on simple, single-stage tasks and lack the adaptability required for multilevel GCPs. Specifically, DMs often lack the ability to dynamically coordinate decisions across levels, and their generative nature makes it difficult to guide solutions toward problem-specific objectives Sanokowski et al. (2024a); Du et al. (2024a;b).

To bridge these gaps, we propose a reinforcement learning-guided diffusion model (RLG-DM) tailored for multilevel GCPs. RLG-DM integrates DM with reinforcement learning (RL)-based control to enable progressively scalable, controllable solution generation. Specifically, the forward diffusion process is used to learn transferable structural priors from simpler sub-GCPs using GNNs. During inference, a reinforcement learning controller is embedded in the reverse diffusion process, interacting with intermediate graph states to hierarchically guide solution refinement toward objectives such as feasibility, cost, or constraint satisfaction. The contributions of this study are summarized below:

- We propose a unified RL-based graph diffusion model framework (RLG-DM) for multilevel GCPs, where the information gain-based adaptive noise scheduling mechanism improves structural robustness during the forward diffusion process, and the multilevel Q-value aggregation scheme enables multilevel coordination during the reverse diffusion process, collectively supporting robust generalization in multilevel GCPs.

- RLG-DM departs from conventional diffusion models by embedding reinforcement feedback into the reverse diffusion trajectory. This design transforms the process from passive denoising into an active, reward-aligned reasoning mechanism, enabling coherent and task-relevant decision-making across multilevel graph structures.

- Extensive experiments on representative multilevel GCPs, including LRP and NRP, demonstrate that RLG-DM outperforms state-of-the-art methods and generalizes well across structurally diverse tasks.

## 2 PRELIMINARY

### 2.1 GRAPH COMBINATORIAL PROBLEMS

GCPs identify the optimal subgraphs within a graph $G = (V, E)$ that meet predefined performance standards and constraints. These problems are prevalent across multiple domains, including logistics Yan et al. (2023), communication networks Sun et al. (2024), and social systems Wang (2018). A GCP is formulated on a graph $G = (V, E)$ with the objective of selecting a subset of vertices $V' \subseteq V$ and edges $E' \subseteq E$ such that the resulting subgraph $G' = (V', E')$ optimizes a given objective function while satisfying a set of constraints.

$$\text{Max (or Min)} \quad f(V', E') \qquad \text{s.t} \quad C(V', E') \tag{1}$$

where $f$ is the objective function and $C$ represents the constraints. Multilevel GCPs extend the GCPs by incorporating multiple levels of hierarchy within the graph structure. These problems involve not only identifying optimal subgraphs but also considering the interrelations between different multilevel layers. In a multilevel GCPs, the graph $G$ is structured into several layers, where each layer $L_k$ represents a different level of the hierarchy, and $K$ denotes the number of multilevel layers:

$$G = \bigcup_{k=1}^{K} L_k, \quad L_k = (V_k, E_k) \tag{2}$$

The objective is to select subgraphs $G'_k = (V'_k, E'_k)$ from each layer such that the overall performance is optimized across all layers. The detailed mathematical formulation of multilevel GCPs is provided in Appendix A.8.

The main challenges of multilevel GCPs include handling the computational demands and achieving overall optimality across multiple layers Peng et al. (2021). A representative example is the LRP, which can be viewed as a two-level hierarchical GCP, with a Facility Location Problem (FLP) at the upper level Snyder (2006) and multiple Vehicle Routing Problems (VRPs) at the lower level Eksioglu et al. (2009). Decisions at each level impact and constrain choices at the other. Each subproblem operates on a distinct graph and interacts with others through multilevel constraints. Similarly, the NSP Strandmark et al. (2020) can be decomposed into a graph coloring problem Wang et al. (2024) and an assignment problem Spivey & Powell (2004), both of which reflect multilevel GCP characteristics. Decisions at the coloring stage influence the complexity and feasibility of the assignment stage, further illustrating multilevel interdependence. These examples collectively highlight the complexity, diversity, and tightly coupled decision-making processes inherent in multilevel GCPs, motivating the development of unified models capable of multilevel reasoning across structured graph representations.

## 2.2 DIFFUSION MODELS

DMs have garnered widespread attention in both research and industry due to their exceptional generative performance. They have achieved significant success in traditional areas such as computer vision Rombach et al. (2022) and speech synthesis Lu et al. (2022), and have recently shown considerable potential in optimization problems Fang et al. (2024). DMs effectively generate new data by simulating the forward diffusion process (gradually introducing noise) and the reverse generation process (gradually removing noise). Taking the classical denoising diffusion probabilistic models (DDPM) Ho et al. (2020) as an example, the forward diffusion process incrementally transforms the original data $x_0$ into noise, represented as a Markov chain. At each time step $t$, the data $x_{t-1}$ undergoes the following probability distribution to add noise:

$$x_t = \sqrt{1 - \beta_t} x_{t-1} + \sqrt{\beta_t} \epsilon, \quad \epsilon \sim \mathcal{N}(0, I) \tag{3}$$

where $\beta_t$ is the noise level at time step $t$, and $\epsilon$ is noise sampled from a standard normal distribution. The reverse generation process is the inverse of the forward process, aimed at recovering original data from noisy data. This process can gradually reduce noise and restore data through the following probability distribution:

$$x_{t-1} = \frac{1}{\sqrt{\alpha_t}} \left( x_t - \frac{\beta_t}{\sqrt{1 - \bar{\alpha}_t}} \epsilon_\theta(x_t, t) \right) \tag{4}$$

Here, $\bar{\alpha} = \prod_{j=1}^{t} \alpha_j$ and $\alpha_t = 1 - \beta_t$ is a variance schedule. $\epsilon_\theta(x_t, t)$ is the noise component predicted by a neural network based on the current noisy data $x_t$ and time step $t$, used to progressively restore the data. The model is trained by minimizing the variational lower bound between the forward and reverse trajectories, ensuring faithful data reconstruction. Although originally developed for continuous domains, DMs have recently been extended to address discrete optimization.

## 3 PROPOSED METHOD

### 3.1 OVERVIEW

We illustrate the overall framework of the RLG-DM for multilevel GCPs in Fig. 1. RLG-DM operates in two stages: a forward diffusion stage and a reverse diffusion stage. In the forward stage, diffusion models are trained on structurally simpler sub-GCPs, e.g., facility location problem (FLP), vehicle routing problem (VRP), graph coloring problems (GC), assignment problem (AP), and traveling salesman problem (TSP), using GNN-based encoders guided by information gain, enabling the learning of transferable structural priors through progressive noise injection. In the reverse stage, these learned priors are reused and composed to tackle multi-level GCPs. An RL controller is integrated to guide the iterative denoising process, steering the generation of solutions toward feasibility and task-specific optimality. Training proceeds in two phases: first, GNN encoders are pretrained on progressively noised sub-GCPs to learn robust structural representations; second, an RL controller is trained to compose these frozen encoders for multilevel denoising. During inference, the pretrained components work together to reconstruct multilevel solutions from noise, enabling generalization without retraining. The reverse diffusion process is formalized as an Markov Decision Process (MDP) (Section 3.3.1), with a reward function that balances constraint satisfaction and problem-specific objective optimization.

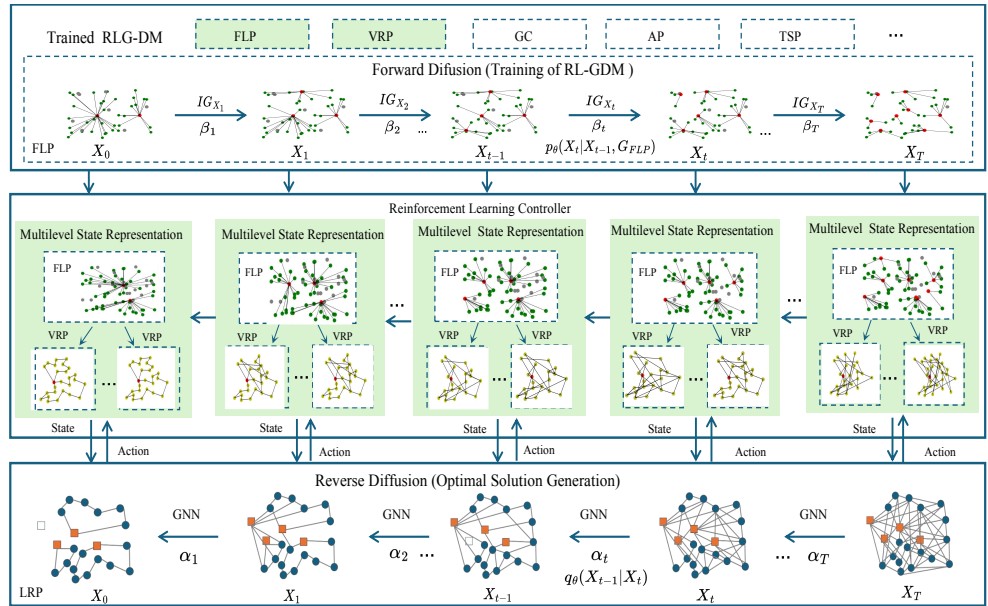

Figure 1: The framework of our RLG-DM. In this example, the multilevel GCP is LRP, with sub-GCPs as FLP and VRP. The forward diffusion process trains the model by using GNNs to optimize information gain for different GCPs. The reverse diffusion process generates optimal solutions via a reinforcement learning controller based on the trained RLG-DM.

## 3.2 FORWARD DIFFUSION WITH GNN ENCODING

We train GNN-based encoders using a forward diffusion process that progressively injects noise into graph structures, pushing node representations away from optimal configurations. As described in Algorithm 1 in Appendix A.1, the DM is trained on solved instances of various sub-GCPs $(G, X_0)$, such as facility location and vehicle routing, with each sub-GCP being trained on its own distinct set of instances. To balance exploration and structural preservation, an adaptive noise schedule modulates the diffusion intensity $\sigma_t$ at each step based on information gain. This mechanism enables RLG-DM to acquire transferable structural priors that generalize effectively to multilevel GCPs.

### 3.2.1 GRAPH-BASED LEARNING DIFFUSION

In this section, we distinguish between graph-level configuration and node-level embeddings. Let $X_t$ represent the graph configuration (node features and connectivity), and $\mathbf{h}_i^t \in \mathbb{R}^d$ denote the embedding of node $i$ produced by the GNN. Noise is injected during GNN message passing (Eq. (5)), perturbing the learned representations while preserving the underlying graph structure. Let $G = (V, E)$ denote a graph representing a GCP instance, where $V$ is the node set and $E$ the edge set. The embedding of node $i \in V$ in diffusion step $t$ is $\mathbf{h}_i^t = [h_{i1}^t, \ldots, h_{id}^t] \in \mathbb{R}^d$. Node embeddings evolve according to the following:

$$\mathbf{h}_i^t = \sigma \left( \mathbf{W} \left( \sum_{j \in \mathcal{N}(i)} \mathbf{h}_j^{t-1} + \epsilon_i^t \right) + \mathbf{b} \right) \tag{5}$$

where $\mathcal{N}(i)$ denotes neighbors of node $i$, $\mathbf{W}$ and $\mathbf{b}$ are trainable GNN parameters, $\sigma(\cdot)$ is ReLU activation, and $\epsilon_i^t \sim \mathcal{N}(0, \sigma_t^2)$ is Gaussian noise injected during aggregation. This formulation enables the model to learn robust representations under progressive perturbation.

### 3.2.2 ADAPTIVE NOISE SCHEDULING VIA INFORMATION GAIN

To balance exploration and stability during diffusion, we introduce an adaptive noise scheduling strategy based on information gain, which measures the reduction in node-level entropy over diffu-

sion steps:

$$IG(X_t) = \sum_{i \in V} w_i \cdot \left(H(\mathbf{h}_i^{t-1}) - H(\mathbf{h}_i^t)\right) \tag{6}$$

where $w_i = \frac{d_i}{\sum_{j \in V} d_j}$ is the normalized degree-based importance of node $i$, and $H(\mathbf{h}_i^t) = -\sum_{k=1}^{d} p_k^t(i) \log p_k^t(i)$ with $p_k^t(i) = \frac{e^{h_{ik}^t}}{\sum_{j=1}^{d} e^{h_{ij}^t}}$. Decreasing $IG(X_t)$ over successive diffusion steps indicates increased uncertainty, reflecting effective noise injection.

Based on $|IG(X_t)|$, we dynamically switch between three noise schedules: linear ($\sigma_t = \sigma_{\min} + \frac{t}{T}(\sigma_{\max} - \sigma_{\min})$, baseline), cosine ($\sigma_t = \sigma_{\max} \cdot \cos(\frac{\pi t}{2T})$, for moderate gain), and exponential ($\sigma_t = \sigma_{\min} \cdot (\frac{\sigma_{\max}}{\sigma_{\min}})^{\frac{t}{T}}$, for aggressive exploration). We initialize with linear and switch to cosine when $0.05 < |IG(X_t)| \leq 0.1$, or exponential when $|IG(X_t)| > 0.1$.

### 3.2.3 TRAINING OF DIFFUSION-CONDITIONED GNNs

Unlike traditional GNNs that operate on static graphs, our model learns node representations that remain robust under increasing levels of stochastic noise introduced by the forward diffusion process. To this end, we define a training objective that encourages both accurate reconstruction and temporal consistency across diffusion steps. The loss at each step $t$ is formulated as:

$$\mathcal{L}_t = \|\mathbf{h}_t^{\text{target}} - \mathbf{h}_t\|^2 + \mu\|\mathbf{h}_t - \mathbf{h}_{t-1}\|^2 \tag{7}$$

where $\mathbf{h}_t$ is the noisy node embedding at time step $t$, $\mu$ is a regularization hyperparameter, and $\mathbf{h}_t^{\text{target}}$ denotes the denoising target. In practice, we set $\mathbf{h}_t^{\text{target}} = \mathbf{h}_0$, i.e., the clean embedding at the beginning of the diffusion process. The first term in Eq. (7) serves as a reconstruction loss, encouraging the model to recover clean features from noisy inputs. The second term acts as a temporal smoothness regularizer, penalizing abrupt shifts between successive steps. This helps maintain structural coherence and improves training stability under high noise conditions.

We train one GNN encoder per sub-GCP type independently on their respective benchmark datasets. For example, the FLP encoder is trained on solved FLP instances, while the VRP encoder is trained on solved VRP instances. This parallel training strategy leverages existing benchmarks and enables modular pretraining, which allows new sub-GCP encoders to be added without retraining existing components. During reverse diffusion, these pretrained encoders are frozen and combined to handle multilevel problems. Crucially, by training GNNs on progressively noised graphs during forward diffusion, the encoders learn robust representations that remain valid even when the graph structure is dynamically modified by the RL agent during the reverse process. This aligns the static pretraining with the dynamic inference environment.

### 3.3 REINFORCEMENT-GUIDED REVERSE DIFFUSION

In RLG-DM, reverse diffusion is an iterative, reinforcement-guided denoising process that incrementally refines a noise-injected graph toward a feasible and task-optimal solution. Unlike classical DMs that rely on analytically derived score functions or learned stochastic transitions, we cast the reverse process as a sequential decision-making problem formalized as a Markov Decision Process (MDP). At each time step $t$, the RL agent observes the current multilevel graph state and selects actions to reduce structural noise while improving solution quality. The agent is guided by an RL controller trained on structural priors acquired during the forward diffusion stage. It operates over a multilevel decomposition of the original GCP, organized into multiple layers of sub-GCPs.

### 3.3.1 MDP FORMULATION

We formalize the reverse diffusion process as an MDP defined by the tuple $(S, \mathcal{A}, P, R, \gamma)$. The state space $S$ consists of multilevel graph configurations. For $L$ levels with $M_l$ subproblems at level $l$, state $s_t$ is constructed using frozen GNN encoders from forward diffusion. For each subproblem $i$ at level $l$:

$$\mathbf{H}_{l,i}^t = \text{GNN}_{l,i}(\mathbf{A}_{l,i}^t, \mathbf{H}_{l-1}^t), \quad s_t = \left[\mathbf{H}_{1,1}^t, \ldots, \mathbf{H}_{L,M_L}^t\right] \tag{8}$$

where $\mathbf{A}_{l,i}^t$ is the adjacency matrix and $\mathbf{H}_{l-1}^t$ denotes embeddings from the previous level. Sub-GCPs at the same level share one encoder, enabling modular reuse. The action space $\mathcal{A}$ consists

of discrete graph modifications tailored to each sub-GCP type. For LRP: Level 1 (FLP) actions open/close facilities; Level 2 (VRP) actions insert customers into routes or swap edges. Each action modifies $\mathbf{A}_{l,i}^t$, inducing transition $X_t \rightarrow X_{t-1}$ (detailed action specifications are provided in Appendix A.9.1). The transition function $P(s_{t+1}|s_t, a_t)$ is deterministic. Although actions modify graph structure, the frozen GNN encoders remain effective because: (1) they were trained on progressively noised graphs during forward diffusion, learning robust features under perturbations; (2) they encode local relational patterns resilient to incremental modifications; and (3) the denoising process refines structure gradually, avoiding drastic changes that could invalidate learned embeddings. The reward function $R$ balances cost minimization satisfying the following constraint:

$$r_t = -C(s_t, a_t) - \lambda \cdot V(s_t, a_t), \tag{9}$$

where $C(s_t, a_t)$ is the total cost, $V(s_t, a_t)$ quantifies constraint violations, and $\lambda \in [3.0, 10.0]$ is a penalty coefficient tuned per problem to balance cost and feasibility. We set the discount factor $\gamma = 0.99$ for long-term planning.

### 3.3.2 JOINT ACTION SELECTION VIA MULTILEVEL Q-AGGREGATION

At each time step $t$, the agent selects an action using a value-based strategy. Each sub-GCP $i$ at level $l$ maintains its own Q-function $Q_{l,i}(s, a)$, implemented as a deep Q-network (DQN). Implementation details including the network architecture, experience replay and target network updates, are provided in Appendix A.9.2. For levels with multiple subproblems, we first compute the average Q-value:

$$Q_l(s, a) = \frac{1}{M_l} \sum_{i=1}^{M_l} Q_{l,i}(s, a) \tag{10}$$

and then aggregate layer-level Q-values into a joint estimator:

$$Q_{\text{joint}}(s_t, a) = \sum_{l=1}^{L} \varsigma_l \cdot Q_l(s_t, a) \tag{11}$$

The weights $\varsigma_l$ are learnable parameters optimized end-to-end during training. They are initialized uniformly ($\varsigma_l = 1/L$) and updated via gradient descent based on the joint Q-learning loss derived from $Q_{\text{joint}}$. This allows the model to dynamically adapt to task-specific hierarchies. For example, in LRP, the FLP (Level 1) may dominate early training, while VRP agents (Level 2) become more critical as routing details are refined. The agent selects actions via $a_t = \arg\max_a Q_{\text{joint}}(s_t, a)$ using $\varepsilon$-greedy exploration during training. Each level Q-function is updated via the standard Bellman equation:

$$Q_{l,i}(s_t, a_t) \leftarrow Q_{l,i}(s_t, a_t) + \eta \Big( r_t + \gamma \max_{a'} Q_{l,i}(s_{t+1}, a') - Q_{l,i}(s_t, a_t) \Big) \tag{12}$$

where $\eta$ is the learning rate. The complete training procedure is detailed in Algorithm 2 in Appendix A.10. This multilevel aggregation scheme enables globally coherent action selection while preserving the granularity of local sub-GCP decision-making.

### 3.3.3 INFERENCE FOR OPTIMAL SOLUTION GENERATION

At inference time, RLG-DM reuses the pretrained GNN encoders and learned RL controller without further optimization. The process starts from a fully noise-injected graph $X_T$ and iteratively refines it toward a solution. At each step $t$, the agent encodes the current state via frozen GNNs, aggregates Q-values across levels (Eq. 10-11), selects the optimal action $a_t = \arg\max_a Q_{\text{joint}}(s_t, a)$, and applies it to progressively denoise the structure. The complete inference procedure is provided in Algorithm 3 in Appendix A.10. Although the hierarchy involves multiple levels and subproblems, GNN encoding and Q-value aggregation operations are vectorized and executed in parallel, enabling efficient inference. The overall inference complexity is $O(T \times L)$, where $T$ is the number of diffusion steps and $L$ is the number of hierarchical levels. Since $L$ is typically small (2-3 for most multilevel GCPs), the complexity scales near-linearly with diffusion steps, effectively avoiding the computational overhead of naive nested loops. Although the penalized reward (Eq. 9) strongly encourages validity, stochastic generation may yield minor violations. To ensure strict satisfaction of hard constraints, we apply a lightweight deterministic repair operator $\mathcal{R}(\cdot)$ to the final output $X_0$. For example, in LRP, unserved customers are greedily inserted into the nearest feasible route. See Appendix A.2 for detailed analysis.

## 4 EXPERIMENTAL RESULTS

We conduct comprehensive experiments to evaluate RLG-DM regarding (1) the solution quality compared to state-of-the-art baselines; (2) the generalization capability across problems with varying scales; and (3) the ablation analysis to validate the contribution of key components. We evaluate RLG-DM on three diverse multilevel GCPs: LRP, which merges FLP and VRP for spatial optimization; NSP, which coordinates shift assignment (via GC) and workload optimization (via AP); and the Flexible Job Shop Scheduling Problem (FJSP), which combines AP, TSP, and GC for production scheduling. Detailed problem formulations and additional results, including FJSP, multi-echelon LRP variants for hierarchical scalability, and computational efficiency analysis, are provided in Appendix A.1.

### 4.1 EXPERIMENTAL SETTINGS

We evaluate on the LRP dataset[1] Prins et al. (2006), the NSP dataset[2]Curtois & Qu (2014) and FJSP dataset[3]. The LRP dataset comprises 12 instances with 100 or 200 customers and 10 depots, varying in capacity constraints and cost structures (suffixes -1/-2/-3 and a/b variants). The NSP dataset contains 24 benchmark instances differing in planning periods (2-52 weeks), workforce sizes (8-150 employees), and shift type complexities (1-32 types). We evaluated all 12 LRP instances and 12 selected NSP instances (Instances 10-21) that systematically cover planning horizons from 4 to 26 weeks, workforce sizes from 20 to 120 employees, and shift complexities from 3 to 18 types.The FJSP dataset contains 80 instances divided into 8 groups, with each group varying in the number of jobs and machines. Our RLG-DM is trained on simpler GCPs to learn general combinatorial optimization patterns. Specifically, we use FLP and AP from the OR-Library[4], VRP from the Solomon benchmark[5], GC from the COLOR datasety[6], and TSP from the TSPLIB dataset Reinelt (1991). This progressive training strategy enables the model to learn transferable optimization strategies that generalize effectively to complex multilevel problems.

We conducted a comprehensive comparison of our RLG-DM with five state-of-the-art approaches: DIFUSCO Sun & Yang (2023), T2T Li et al. (2024), RLCA Delarue et al. (2020), MEO-HFG Yan et al. (2023), and COMBHELPER Tian et al. (2024). DIFUSCO and T2T are diffusion model (DM)-based solvers, COMBHELPER utilizes a GNN-based approach, RLCA is based on reinforcement learning, and MEO-HFG employs a graph-based evolutionary algorithm Yan et al. (2023). For fair comparisons, we utilized the recommended parameter settings for all baseline algorithms as reported in their respective literature.

For RLG-DM, the number of training epochs is set to 100, and the batch size to 64. Network parameters include a hidden layer size of 64, a learning rate of 0.001, ReLU activation, Adam optimizer, and a dropout rate of 0.5. The GNN model, specifically GraphSAGE Hamilton et al. (2017), features 2 graph convolutional layers with node feature dimensions of 64 and edge feature dimensions of 32. The number of diffusion steps is fixed at 1000 to ensure stable optimization. All experiments were conducted on an NVIDIA L40 GPU.

### 4.2 RESULTS ON THE LRPS

Table 1 summarizes the comparative results on LRP instances. We report the average cost (Avg) and standard deviation (Std) over 30 independent runs. To strictly validate the performance gaps, we conduct the Wilcoxon rank-sum test, with $p$-values indicating the statistical significance of the improvement compared to the best baseline. As shown in Table 1, RLG-DM achieves the lowest average cost on 8 out of 12 instances. The statistical analysis confirms that these improvements are robust. The $p$-values are well below 0.05 (e.g., 0.001 or 0.008) for the majority of the winning instances, indicating that RLG-DM's superiority is statistically significant and not due to random variance. In terms of pairwise comparison (see the w/d/l row), RLG-DM dominates RLCA, DIFUSCO, and T2T, outperforming them on 11 out of 12 instances. Against the strongest competitor, MEO-HFG, our method still maintains a clear lead, winning on 9 instances. The results highlight the scalability of RLG-DM. While the performance is competitive on smaller 100-customer instances, RLG-DM demonstrates distinct advantages as problem complexity increases. Specifically, on the

---

[1]http://prodhonc.free.fr/Instances/instances_us.htm;    [2]http://www.schedulingbenchmarks.org/nrp/;
[3]https://optimizizer.com/TA.php;    [4]http://people.brunel.ac.uk/ mastjjb/jeb/info.html;
[5]https://www.sintef.no/projectweb/top/vrptw/solomon-benchmark/;    [6]https://mat.tepper.cmu.edu/COLOR02/

Table 1: Performance comparison on LRP benchmark

| Instances | RLCA | | MEO-HFG | | COMBHELPER | | DIFUSCO | | T2T | | RLG-DM | | p-value |
|---|---|---|---|---|---|---|---|---|---|---|---|---|---|
| | Avg ↓ | Std ↓ | Avg ↓ | Std ↓ | Avg ↓ | Std ↓ | Avg ↓ | Std ↓ | Avg ↓ | Std ↓ | Avg ↓ | Std ↓ | |
| 100-10-1a | 321,031 | 121.32 | 293,046 | 189.21 | 298,328 | 145.21 | 301,021 | 102.34 | 298,341 | 103.21 | **291,421** | 103.24 | 0.008 |
| 100-10-1b | 289,012 | 59.23 | 246,067 | 92.34 | **234,646** | 98.23 | 299,021 | 123.29 | 239,310 | 89.01 | 234,921 | 87.23 | - |
| 100-10-2a | 267,041 | 89.34 | 250,476 | 69.34 | 252,312 | 103.31 | 268,132 | 74.19 | 249,195 | 96.81 | **245,192** | 68.93 | 0.000 |
| 100-10-2b | 230,124 | 77.31 | **210,934** | 78.23 | 214,061 | 80.32 | 230,213 | 83.12 | 287,122 | 78.12 | 214,912 | 108.23 | - |
| 100-10-3a | 259,021 | 108.23 | 256,075 | 93.79 | 261,046 | 32.12 | 259,213 | 76.20 | **255,243** | 60.12 | 261,219 | 76.62 | - |
| 100-10-3b | 219,463 | 49.91 | 210,800 | 85.91 | 221,219 | 87.34 | 221,321 | 89.19 | 219,391 | 31.90 | **210,712** | 49.12 | 0.050 |
| 200-10-1a | 493,321 | 180.29 | 480,762 | 189.49 | 482,481 | 56.01 | 491,230 | 72.49 | 483,415 | 68.31 | **480,301** | 56.23 | 0.040 |
| 200-10-1b | 393,398 | 101.34 | 385,216 | 102.31 | 391,412 | 87.30 | 394,521 | 89.13 | 381,415 | 40.21 | **380,731** | 87.20 | 0.042 |
| 200-10-2a | 459,321 | 86.38 | 455,156 | 103.41 | 460,218 | 101.32 | 461,314 | 94.29 | 460,281 | 58.38 | **451,901** | 91.32 | 0.001 |
| 200-10-2b | 387,319 | 181.27 | **379,929** | 178.02 | 389,219 | 102.91 | 382,301 | 101.20 | 381,919 | 62.19 | 381,516 | 82.10 | - |
| 200-10-3a | 483,241 | 138.19 | 484,156 | 183.21 | 490,139 | 109.18 | 484,313 | 98.20 | 480,123 | 90.18 | **474,801** | 69.32 | 0.000 |
| 200-10-3b | 378,312 | 121.41 | 378,556 | 186.67 | 379,421 | 70.31 | 380,122 | 80.19 | 381,290 | 70.19 | **365,210** | 98.12 | 0.000 |
| mean | 348,384 | 109.52 | 335,931 | 129.33 | 339,542 | 89.46 | 347,727 | 90.32 | 343,087 | 70.72 | **332,736** | 81.47 | |
| w/d/l | 1/0/11 | | 3/0/9 | | 3/0/9 | | 1/0/11 | | 1/0/11 | | | | |

**Notes:** Values represent Avg and Std over 30 runs. **Bold** and underlined values indicate the best and second-best results.
'w/d/l' denotes the number of wins, draws, and losses of the baseline algorithm compared to RLG-DM.
p-values compare RLG-DM against the best baseline using the Wilcoxon rank-sum test; '-' indicates no improvement.

Table 2: Comparative results on the NSP benchmark.

| Instances | RLCA | | MEO-HFG | | COMBHELPER | | DIFUSCO | | T2T | | RLG-DM | | p-value |
|---|---|---|---|---|---|---|---|---|---|---|---|---|---|
| | Best ↓ | Gap ↑ | Best ↓ | Gap ↑ | Best ↓ | Gap ↑ | Best ↓ | Gap ↑ | Best ↓ | Gap ↑ | Best ↓ | Gap ↑ | |
| Instance 10 | **4,631** | 79.01% | 4,920 | 77.70% | 4,789 | 78.30% | 4,644 | 78.95% | **4,631** | 79.01% | **4,631** | 79.01% | - |
| Instance 11 | **3,443** | 87.33% | 3,487 | 87.17% | **3,443** | 87.33% | 3,466 | 87.25% | 3,486 | 87.17% | **3,443** | 87.33% | - |
| Instance 12 | 5,601 | 79.22% | 4,087 | 84.84% | **4,040** | 85.01% | 4,712 | 82.52% | 5,211 | 80.67% | **4,040** | 85.01% | - |
| Instance 13 | **2,008** | 94.35% | 4,311 | 87.88% | 3,189 | 91.03% | 2,900 | 91.84% | 3,900 | 89.03% | 2,848 | 91.99% | - |
| Instance 14 | 1,899 | 92.08% | 1,400 | 94.16% | 1,280 | 94.66% | 1,566 | 93.47% | 1,308 | 94.55% | **1,278** | 94.67% | 0.045 |
| Instance 15 | 4,190 | 88.02% | 5,100 | 85.42% | 6,133 | 82.47% | 4,190 | 88.02% | **3,871** | 88.94% | 3,921 | 88.79% | - |
| Instance 16 | 5,769 | 63.51% | 3,852 | 75.63% | 4,321 | 72.67% | 3,789 | 76.03% | **3,225** | 79.60% | 3,225 | 79.60% | - |
| Instance 17 | 5,978 | 82.62% | 6,800 | 80.23% | 6,100 | 82.26% | 5,969 | 82.64% | 5,971 | 82.64% | **5,802** | 83.13% | 0.002 |
| Instance 18 | 4,679 | 83.71% | 7,920 | 72.43% | 4,950 | 82.77% | 4,560 | 84.13% | 5,612 | 80.47% | **4,459** | 84.48% | 0.005 |
| Instance 19 | 5,246 | 89.44% | 4,198 | 91.55% | 3,198 | 93.56% | 4,187 | 91.57% | 3,190 | 93.58% | 3,345 | 93.26% | - |
| Instance 20 | 8,762 | 84.57% | 12,233 | 78.46% | 8,762 | 84.57% | 5,242 | 90.77% | 4,987 | 91.22% | **4,780** | 91.58% | 0.000 |
| Instance 21 | 62,323 | 61.02% | 98,872 | 38.16% | 34,212 | 78.60% | 34,182 | 78.62% | 33,121 | 79.28% | **32,133** | 79.90% | 0.000 |
| Mean Gap | | 82.07% | | 79.47% | | 84.44% | | 85.48% | | 85.51% | | **86.56%** | - |
| w/d/l | 1/2/9 | | 0/0/12 | | 1/2/9 | | 0/0/12 | | 2/2/8 | | | | |

**Notes:** **Bold** and underlined values indicate the best and second-best results.
'w/d/l' denotes the number of wins, draws, and losses of RLG-DM compared to the baseline.
P-values indicate statistical significance compared to the best baseline; '-' indicates no improvement or a tie.

larger 200-customer instances (from 200-10-1a to 200-10-3b), RLG-DM achieves the best known results on 5 out of 6 instances. This suggests that RLG-DM effectively captures complex node relationships in larger graphs, whereas baselines like MEO-HFG struggle to maintain performance as the problem scale doubles.

## 4.3 RESULTS ON THE NSPS

Table 2 summarizes the comparative performance of RLG-DM on NSP instances. To ensure a fair comparison, all algorithms start from the same initial solution generated by randomly assigning nurses to shifts while satisfying hard constraints Strandmark et al. (2020). We report the best cost found over 30 independent runs and the "Gap" (%), which measures the relative improvement from the initial solution. RLG-DM demonstrates a significant advantage over the baselines, achieving the best objective values on 9 out of 12 instances. In terms of solution quality, our method attains the highest average improvement gap of 86.56%, surpassing the strongest baseline T2T (85.34%) and significantly outperforming RLCA (82.07%). As indicated by the $p$-values, RLG-DM achieves statistically significant improvements ($p < 0.05$) on key instances (e.g., Instance 17, 18, 20, 21), confirming that the performance gains are robust and not due to random chance. Compared to other diffusion-based approaches, RLG-DM shows decisive dominance. Specifically, it strictly outperforms DIFUSCO on all 12 instances and outperforms or matches T2T on 8 instances. Notably, on the largest and most complex instance (Instance 21), RLG-DM achieves a cost of 32,133, beating T2T (33,121) by a substantial margin of nearly 1,000 units. The superior performance, particularly on larger instances, validates the effectiveness of the proposed adaptive noise scheduling. By dynamically adjusting the guidance during the reverse diffusion process, RLG-DM can better capture the hierarchical constraints of multilevel GCPs. This capability allows it to escape local optima more effectively than T2T and DIFUSCO, which rely on standard noise schedules.

## 4.4 PERFORMANCE ANALYSIS ACROSS PROBLEM SCALES

Figure 2 demonstrates a robust scalability pattern across both problem types. For LRP, the mean improvement over the best baseline shifts from $-0.36\%$ on 100-customer instances ($n = 6$) to $0.86\%$ on 200-customer instances ($n = 6$). A similar trend is observed in NSP, where the performance gap widens from $0.15\%$ on smaller workforces ($< 50$ employees, $n = 7$) to $1.27\%$ on larger ones ($\geq 50$ employees, $n = 5$). This indicates that while modern baselines remain competitive on less constrained, smaller-scale problems, RLG-DM excels as the solution space becomes more intricate. The adaptive noise scheduling and RL guidance allow RLG-DM to effectively navigate the hierarchical constraints of large-scale tasks, resulting in substantial performance gains as complexity grows. Extensive experiments on FJSP and multi-echelon LRP variants in Appendix A.1 further corroborate this generalization capability.

## 4.5 ABLATION STUDIES

We conduct an ablation study to evaluate the contributions of the key components in our RLG-DM. Experiments are performed on both LRP and NSP, with results averaged over 12 representative instances for each problem. To assess the effectiveness of the adaptive scheduling strategy, we compare our RLG-DM with three fixed diffusion schedule variants: linear (RLG-DM(L)), cosine (RLG-DM(C)), and exponential (RLG-DM(E)). In addition, we introduce a variant, RLG-DM-H, which is trained and tested directly on different instances of the same multilevel GCP, without progressive training on sub-GCPs. As shown in Table 3, our RLG-DM consistently achieves the lowest

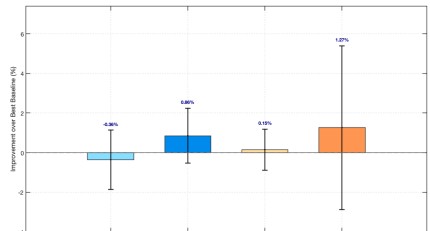

Figure 2: Mean performance improvement of RLG-DM stratified by problem scale. LRP instances grouped by customer count (100 vs. 200); NSP instances grouped by employee count ($< 50$ vs. $\geq 50$).

mean best cost compared to three fixed schedule variants and RLG-DM-H on both 12 LRP and NSP instances. This demonstrates that the noise schedule selection significantly affects the optimization performance, and that dynamically adjusting the schedule according to information entropy provides substantial advantages. Notably, RLG-DM-H performs worse than the RLG-DM, highlighting the effectiveness of the progressive training strategy and the RL-based controller for generalization and robustness. More ablation results are provided in Appendix A.1.4.

## 5 CONCLUSION

In this paper, we introduce a reinforcement learning-guided diffusion model designed to leverage the diffusion process in addressing the scalability challenges inherent in solving multilevel GCPs.In the proposed RLG-DM, adaptive noise scheduling with information gain is used in graph-based forward diffusion, allowing for gradual exploration of a broader range of possible solutions while preserving essential graph structural information for different sub-GCPs. In particular, a reinforcement learning-based control in the reverse diffusion process is

Table 3: Average best costs obtained on the instances of the LRP and NSP problems, respectively.

| Model | LRP ↓ | NSP ↓ |
|---|---|---|
| RLG-DM(L) | 344,529 | 6,601 |
| RLG-DM(C) | 343,104 | 6,832 |
| RLG-DM(E) | 339,450 | 6,709 |
| RLG-DM-H | 335,031 | 13,842 |
| RLG-DM | **333,312** | **6,138** |

employed for joint optimization to generate optimal solutions for multilevel GCPs. Through experiments on LRPs, NSPs, and FJSP, we demonstrate that RLG-DM is a competitive and more efficient multilevel GCP solver compared to state-of-the-art optimization algorithms, even when trained on various sub-GCPs. In the future, we will explore whether RLG-DM can be applied to more complex combinatorial optimization problems, especially those with non-graph-structured subproblems. Additionally, we aim to investigate its extension to real-world applications.

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

# A APPENDIX

## A.1 ADDITIONAL EXPERIMENTAL RESULTS

### A.1.1 LRP, NSP, FJSP PROBLEMS

- **LRP:** The goal of an LRP is to minimize the total cost, which includes both the fixed cost of opening facilities and the variable cost of transportation. The problem is formally defined as minimizing the objective function:

$$\text{Minimize } Z = \sum_{i \in F} f_i x_i + \sum_{(i,j) \in E} c_{ij} y_{ij}, \tag{13}$$

where $f_i$ is the fixed cost of opening facility $i$, $c_{ij}$ is the cost of transporting from facility $i$ to customer $j$, $x_i$ is a binary variable indicating whether facility $i$ is opened, and $y_{ij}$ is a binary variable indicating whether a route between facility $i$ and customer $j$ is used. An LRP typically consists of an FLP in the top level that determines which facilities will be opened, and a set of VRPs in the second level optimizing routes. Crucially, this structure naturally extends to multi-echelon variants (e.g., Two-echelon LRP), where flow cascades through multiple layers of intermediate facilities (satellites). This recursive hierarchy implies that the problem can be scaled by stacking additional assignment and routing layers.

- **NSP:** It is formally defined as minimizing the total cost associated with scheduling, which often includes nurse working hours, overtime pay, and scheduling fairness. The objective function can be expressed as:

$$\text{Minimize } Z = \sum_{n \in N} \sum_{d \in D} (c_{nd} x_{nd} + f_{nd} y_{nd}), \tag{14}$$

where $c_{nd}$ is the basic cost of assigning nurse $n$ to shift $d$, $f_{nd}$ is the overtime cost, $x_{nd}$ is a binary variable indicating whether nurse $n$ is assigned to shift $d$, and $y_{nd}$ is a binary variable indicating whether nurse $n$ is working overtime for shift $d$. An NSP refers to creating schedules for nurses to meet hospital demands while adhering to various constraints. In terms of hierarchy, the graph coloring problem, which represents the assignment of shifts and constraints, ensures that no nurse is assigned to multiple shifts at the same time. The assignment problem, which optimizes specific scheduling and costs, operates on top of this, determines how to assign shifts while minimizing total costs. Thus, the graph coloring problem ensures the legality of the schedules, while the assignment problem optimizes specific scheduling solutions to reduce costs.

- **FJSP:** The Flexible Job Shop Scheduling Problem (FJSP) generalizes the classical job shop setting by introducing routing flexibility, aiming to minimize the makespan $C_{\max}$. The objective function is defined as:

$$\text{Minimize } C_{\max} = \max_{i \in \mathcal{J}} C_i, \tag{15}$$

where $C_i$ denotes the completion time of the last operation of job $i$. The problem involves a set of jobs $\mathcal{J}$ composed of sequential operations and a set of machines $\mathcal{M}$. Each operation can be processed on a subset of eligible machines with varying processing times. Structurally, the FJSP can be decomposed into three interlinked combinatorial layers. First, the AP addresses the routing decision, where operations are allocated to specific eligible machines to balance the workload and reduce potential bottlenecks. Once operations are assigned, the problem on each machine transforms into a sequencing task analogous to the TSP, where the machine acts as a server traversing a "path" of operations to minimize local idle and completion times. Finally, the feasibility of the entire schedule is governed by the GC, which models the conflict resolution; nodes represent operations, and edges enforce strict temporal constraints (job precedence and machine disjunctions) to ensure no two operations overlap on the same resource.

Table 4: Performance comparison on FJSP benchmark

| Method | 15×15 Obj. | Gap | 20×15 Obj. | Gap | 20×20 Obj. | Gap | 30×15 Obj. | Gap | 30×20 Obj. | Gap | 50×15 Obj. | Gap | 50×20 Obj. | Gap | 100×20 Obj. | Gap |
|---|---|---|---|---|---|---|---|---|---|---|---|---|---|---|---|---|
| RLCA | 1531.2 | 24.60 | 1633.5 | 19.43 | 1914.3 | 17.90 | 2151.7 | 21.51 | 2395.4 | 24.13 | 3210.8 | 16.17 | 3325.9 | 16.72 | 5972.6 | 11.59 |
| MEO-HFG | 1519.9 | 23.68 | 1617.0 | 18.22 | 1888.6 | 16.31 | 2105.6 | 18.91 | 2368.5 | 22.73 | 3113.6 | 12.66 | 3246.6 | 13.94 | 5643.4 | 5.43 |
| COMBHELPER | 1425.4 | 15.99 | 1845.9 | 34.95 | 2123.4 | 30.78 | 2037.9 | 15.08 | 2369.3 | 22.77 | 3328.0 | 20.41 | 3545.6 | 24.43 | 6179.4 | 15.45 |
| DIFUSCO | 1447.0 | 17.75 | 1663.1 | 21.59 | 1975.3 | 21.65 | 2218.8 | 25.30 | 2510.0 | 30.07 | 3425.9 | 23.96 | 3641.6 | 27.80 | 6183.9 | 15.53 |
| T2T | **1406.1** | **14.41** | 1624.1 | 18.74 | 1880.8 | 15.83 | 2021.1 | 14.13 | 2347.4 | 21.64 | 3132.0 | 13.32 | 3017.7 | 5.91 | 5655.0 | 5.65 |
| RLG-DM | 1424.8 | 15.93 | **1482.1** | **8.36** | **1688.5** | **3.99** | **1880.0** | **6.17** | **2284.2** | **18.36** | **2908.7** | **5.24** | **2919.8** | **2.47** | **5482.1** | **2.42** |

**Notes:** 80 instances, 8 groups of 10 instances each.

Obj.: makespan ($C_{\max}$) averaged over 10 instances; Gap: percentage gap to theoretical lower bounds (LB).

**Bold** and underlined indicate the best and second-best results, respectively.

All best results by RLG-DM are statistically significant at $p < 0.05$ (Wilcoxon rank-sum test).

Table 5: Performance comparison on two-echelon LRP benchmark

| Instances | RLCA Avg ↓ | Std ↓ | MEO-HFG Avg ↓ | Std ↓ | COMBHELPER Avg ↓ | Std ↓ | DIFUSCO Avg ↓ | Std ↓ | T2T Avg ↓ | Std ↓ | RLG-DM Avg ↓ | Std ↓ | p-value |
|---|---|---|---|---|---|---|---|---|---|---|---|---|---|
| 100-10-8-1 | 393,046 | 189.21 | 393,730 | 90.23 | 395,008 | 220.23 | 425,534 | 230.18 | 410,342 | 240.01 | **392,510** | 92.32 | 0.003 |
| 100-10-8-2 | 446,067 | 92.34 | 413,565 | 83.20 | 410,091 | 100.12 | 485,124 | 110.90 | 465,934 | 120.98 | **402,512** | 70.12 | 0.021 |
| 100-10-8-3 | 450,476 | 69.34 | 347,507 | 72.90 | 375,021 | 80.21 | 391,067 | 85.23 | **270,120** | 90.01 | 386,024 | 68.23 | - |
| 100-10-8-4 | 410,934 | 78.23 | 392,867 | 57.20 | 425,092 | 90.87 | 425,092 | 95.12 | 420,421 | 100.09 | **391,012** | 160.23 | 0.004 |
| 100-10-8-5 | 396,075 | 93.79 | 391,591 | 87.30 | 395,876 | 105.12 | 408,091 | 110.01 | **390,345** | 115.90 | 396,745 | 73.03 | - |
| 200-10-8-6 | 410,871 | 85.49 | 410,428 | 69.39 | 415,512 | 92.10 | 430,500 | 95.30 | 425,501 | 101.21 | **409,671** | 80.21 | 0.001 |
| 200-10-8-7 | 680,762 | 189.49 | **601,463** | 78.32 | 609,120 | 210.32 | 615,219 | 220.21 | 610,010 | 230.21 | 611,324 | 190.81 | - |
| 200-10-8-8 | 641,216 | 102.31 | 642,578 | 99.28 | 695,012 | 110.91 | 640,219 | 115.21 | **605,012** | 120.91 | 641,121 | 110.19 | - |
| 200-10-8-9 | 655,156 | 103.41 | 645,219 | 94.41 | 665,121 | 110.90 | 680,121 | 115.20 | 645,651 | 120.70 | **645,201** | 115.19 | 0.001 |
| 200-10-8-10 | 679,929 | 178.02 | 668,623 | 163.34 | 685,189 | 190.11 | 700,212 | 200.50 | 668,901 | 210.09 | **667,179** | 185.32 | 0.000 |
| 200-10-8-11 | 684,156 | 183.21 | 645,229 | 141.49 | 690,121 | 190.08 | 645,671 | 201.22 | 650,211 | 210.00 | **644,113** | 190.21 | 0.000 |
| 200-10-8-12 | 678,556 | 186.67 | 645,589 | 191.92 | 684,011 | 190.00 | **399,000** | 200.10 | 694,121 | 210.01 | 651,328 | 180.00 | - |
| mean | 543,937 | 129.29 | **516,532** | 102.42 | 537,098 | 140.91 | 521,761 | 148.27 | 521,381 | 155.84 | 519,895 | 126.32 | |
| w/d/l | 1/0/11 | | 4/0/8 | | 2/0/10 | | 4/0/8 | | 4/0/8 | | 7/0/5 | | |

**Notes:** Values represent Avg and Std over 30 runs. **Bold** and underlined values indicate the best and second-best results.

'w/d/l' denotes the number of wins, draws, and losses of the baseline algorithm compared to RLG-DM.

p-values compare RLG-DM against the best baseline using the Wilcoxon rank-sum test; '-' indicates no improvement.

### A.1.2 RESULTS ON THE FJSPs

Table 4 presents results on the FJSP benchmark, which comprises 80 instances organized into 8 groups based on problem size. RLG-DM achieves the best performance on 7 out of 8 instance groups, demonstrating strong effectiveness across varying scales. Notably, RLG-DM exhibits substantial advantages on larger and more complex instances. For the 50×15 group, our method achieves an average gap of 5.24%, significantly outperforming the next best baseline MEO-HFG (12.66%). Similarly, on the largest 100×20 instances, RLG-DM obtains an objective of 5482.1 with merely a 2.42% gap, strictly dominating the best baseline MEO-HFG (5643.4, 5.43% gap). These improvements are statistically significant ($p < 0.05$) across all winning groups. Regarding method comparisons, pure RL-based methods like RLCA generally struggle to maintain feasibility and optimality as constraints tighten. While T2T achieves marginally better results on smaller instances (15×15), indicating that standard diffusion models are competitive on simpler structures, they suffer from scalability issues. In contrast, RLG-DM's consistent dominance on large-scale instances highlights the advantages of our hierarchical decomposition mechanism over both pure RL and flat diffusion approaches.

### A.1.3 HIERARCHICAL SCALABILITY

To further evaluate the generalization capability of RLG-DM on complex hierarchical structures, we extended our evaluation to two multi-echelon variants of the LRP: the two-echelon LRP and the three-echelon LRP. Tables 5 and 6 present the comprehensive results. On the two-echelon LRP (Table 5), RLG-DM achieves the best performance on 7 out of 12 instances. Notably, in terms of overall solution quality, RLG-DM attains the second-lowest mean objective value (519,895), outperforming T2T (521,381) and trailing only MEO-HFG (516,532). While specialized methods like T2T excel on specific instances with highly clustered customer distributions (e.g., 100-10-8-3), RLG-DM demonstrates consistent advantages on instances with balanced hierarchical structures, confirmed by statistically significant improvements ($p < 0.05$) on all winning instances.

Table 6: Performance comparison on three-echelon LRP benchmark

| Instances | RLCA | | MEO-HFG | | COMBHELPER | | DIFUSCO | | T2T | | RLG-DM | | p-value |
|---|---|---|---|---|---|---|---|---|---|---|---|---|---|
| | Avg ↓ | Std ↓ | Avg ↓ | Std ↓ | Avg ↓ | Std ↓ | Avg ↓ | Std ↓ | Avg ↓ | Std ↓ | Avg ↓ | Std ↓ | |
| 100-10-8-5-1 | 605,341 | 182.57 | 592,586 | 174.49 | 598,123 | 188.34 | 587,231 | 195.68 | 607,452 | 203.46 | **583,473** | 168.29 | 0.015 |
| 100-10-8-5-2 | 596,781 | 115.47 | 584,068 | 103.18 | **580,341** | 120.59 | 602,563 | 128.96 | 599,121 | 135.73 | 580,892 | 138.54 | - |
| 100-10-8-5-3 | 620,565 | 98.67 | 607,894 | 89.18 | 614,239 | 105.84 | 626,780 | 112.48 | 603,451 | 118.95 | 599,121 | 85.37 | 0.007 |
| 100-10-8-5-4 | 619,341 | 98.56 | 606,264 | 89.15 | 613,129 | 105.63 | 625,459 | 112.28 | 622,346 | 118.64 | **597,787** | 84.98 | 0.008 |
| 100-10-8-5-5 | 607,235 | 118.38 | 594,071 | 106.18 | 600,892 | 125.49 | 589,561 | 133.97 | 610,127 | 141.26 | **585,451** | 101.85 | 0.011 |
| 100-10-8-5-6 | 618,451 | 142.86 | 605,947 | 129.11 | 602,342 | 150.67 | 624,891 | 159.38 | 621,565 | 167.54 | **597,120** | 123.76 | 0.013 |
| 200-10-8-5-7 | 795,681 | 98.45 | 780,972 | 89.18 | 788,561 | 106.58 | 802,346 | 113.89 | 799,127 | 120.94 | **769,340** | 84.62 | 0.004 |
| 200-10-8-5-8 | 803,453 | 124.36 | 788,341 | 111.87 | 796,121 | 131.69 | 810,560 | 140.27 | 781,237 | 148.35 | **776,891** | 106.93 | 0.006 |
| 200-10-8-5-9 | 854,121 | 113.87 | 839,874 | 102.48 | 833,343 | 120.98 | 861,561 | 129.45 | 858,234 | 136.84 | **827,561** | 97.86 | 0.005 |
| 200-10-8-5-10 | 854,342 | 185.66 | 839,543 | 169.74 | 847,121 | 192.88 | 861,781 | 205.37 | 858,451 | 217.24 | **827,123** | 162.35 | 0.012 |
| 200-10-8-5-11 | 827,563 | 108.97 | 812,289 | 98.19 | **810,341** | 116.23 | 834,781 | 124.65 | 811,458 | 131.96 | 813,450 | 93.64 | - |
| 200-10-8-5-12 | 840,234 | 121.48 | 825,046 | 109.18 | 833,121 | 128.76 | 817,567 | 137.85 | 844,231 | 145.94 | **812,894** | 104.39 | 0.010 |
| mean | 720,259 | 125.78 | 706,408 | 114.33 | 709,806 | 132.81 | 720,423 | 141.19 | 718,067 | 148.90 | **697,592** | 112.72 | |
| w/d/l | 0/0/12 | | 1/0/11 | | 2/0/10 | | 0/0/12 | | 1/0/11 | | | | |

**Notes:** Values represent Avg and Std over 30 runs. **Bold** and underlined values indicate the best and second-best results.
'w/d/l' denotes the number of wins, draws, and losses of the baseline algorithm compared to RLG-DM.
p-values compare RLG-DM against the best baseline using the Wilcoxon rank-sum test; '-' indicates no improvement.

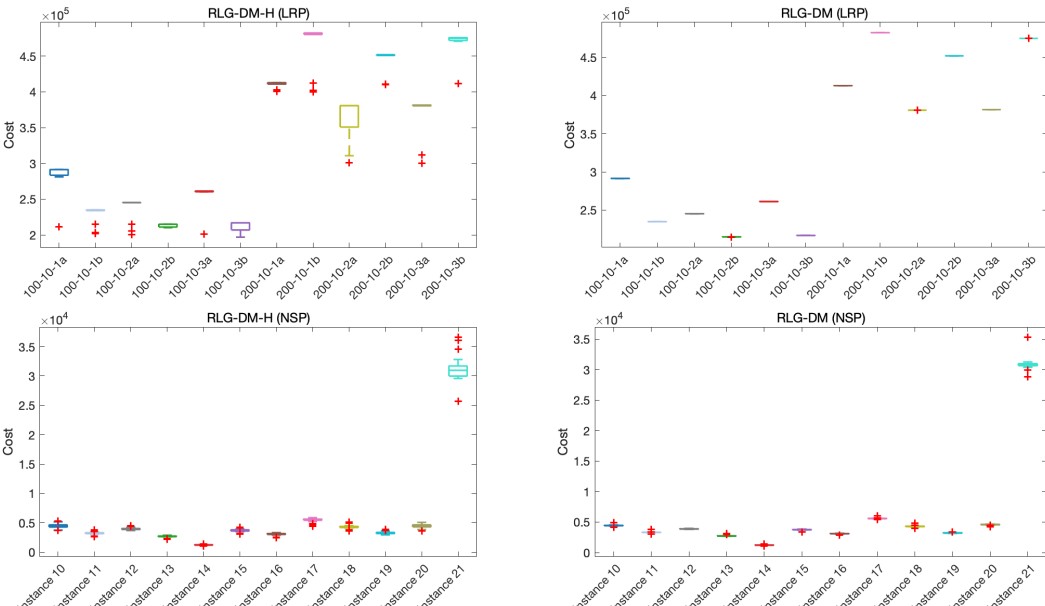

Figure 3: Box plots of the observed cost for LRP and NSP instances, comparing RLG-DM-H and RLG-DM

The results on three-echelon LRP (Table 6) reveal substantially stronger performance, with RLG-DM winning on 10 out of 12 instances and achieving the lowest mean objective value of 697,592. It significantly outperforms all baselines, including MEO-HFG (706,408), COMBHELPER (709,806), and T2T (718,067). The dramatic improvement provides strong evidence that RLG-DM's hierarchical decomposition mechanism scales effectively with problem complexity. Moreover, RLG-DM achieves the lowest standard deviation (112.72) across the three-echelon LRP instances, indicating superior solution stability on deeper hierarchies. Overall, these results validate that RLG-DM demonstrates increasing advantages on more complex hierarchical variants, directly addressing concerns regarding generalization and scalability.

### A.1.4 ADDITIONAL ABLATION STUDIES

Figure 3 further illustrates the boxplots of RLG-DM and RLG-DM-H for LRP and NSP instances, respectively. RLG-DM produces solutions with noticeably smaller cost fluctuations and better stability than RLG-DM-H. This benefit is attributed to the progressive training strategy, where knowledge transfer from FLP and VRP enables more effective multilevel graph structure learning for LRP and NSP.

Table 7: Feasibility analysis of RLG-DM across different multilevel GCPs.

| Task | Total Instances | Raw Feasibility | Instances Requiring Repair |
|------|-----------------|-----------------|----------------------------|
| LRP | 12 | 100% | 0 |
| NSP | 12 | 100% | 0 |
| FJSP | 80 | 96.25% | 3 |
| Two-echelon LRP | 12 | 91.67% | 1 |
| Three-echelon LRP | 12 | 91.67% | 1 |

## A.2 FEASIBILITY ANALYSIS

One major challenge in applying DM to multilevel GCPs is ensuring the feasibility of generated solutions. Prior diffusion-based solvers such as DIFUSCO Sun & Yang (2023) and T2T Li et al. (2024) formulate combinatorial problems as discrete optimization and learn to estimate solution distributions. During inference, these methods rely on post-hoc decoding procedures (e.g., greedy decoding, sampling, or MCTS for DIFUSCO; gradient-based search for T2T) to reconstruct valid discrete solutions. In contrast, RLG-DM explicitly learns to satisfy constraints during the generation process. The RL controller internalizes combinatorial rules through the penalized reward mechanism, allowing the model to generate valid graph topologies directly with minimal dependence on external search procedures.

To evaluate the intrinsic constraint satisfaction capability of our model, we analyzed the raw feasibility rate, defined as the percentage of test instances for which the model generated valid solutions directly without any post-processing. As presented in Table 7, RLG-DM achieves a 100% raw feasibility rate on both standard LRP and NSP datasets, meaning all instances were solved with valid constraints. Even on highly constrained tasks, the model maintains high validity: 96.25% for FJSP (77 out of 80 instances) and 91.67% for multi-echelon LRP variants (22 out of 24 instances). This empirical evidence confirms that the model effectively learns to navigate the constrained solution space via the end-to-end training process.

To provide a strict feasibility guarantee for the rare failure cases, we incorporate deterministic repair operators as lightweight safeguards. Following standard practices in the literature, we adopt the greedy insertion heuristic Dethloff (2002) for LRP to reinsert any unserved customers into feasible routes, the shift-swap procedure Maenhout & Vanhoucke (2011) for NSP to eliminate constraint conflicts, and the left-shift decoding rule Zhang et al. (2025) for FJSP to resolve precedence violations. Since the average raw feasibility is approximately 96% across all benchmarks (with a minimum of 91.67%), these repair steps are triggered in very few instances with negligible computational overhead.

## A.3 SENSITIVITY TO SUB-GCP SELECTION

Table 8 shows that sub-GCP selection significantly impacts RLG-DM's performance, with improper configurations degrading results by 3% to 9%. The aligned sub-GCPs achieve the best performance across both problem scales, with the performance gap widening on larger instances, indicating that proper selection becomes increasingly critical as complexity scales.

The misaligned configuration, which uses structurally unrelated sub-GCPs from NSP (GC and AP designed for discrete staffing tasks), suffers the most severe degradation at 8.67% on average, reaching 9.37% on large instances. This substantial gap demonstrates that sub-GCPs must share core optimization patterns with the target problem to enable positive transfer.

Skipping intermediate stages while maintaining structural alignment (Partial sub-GCPs) degrades performance by 2.81% on average, showing that gradual complexity progression aids learning even when the initial sub-GCP is relevant. The simplified configuration, using only toy problems with 10 to 20 customers, degrades by 7.26% on average, with the gap widening to 7.89% on large instances, confirming that sub-GCPs must be sufficiently complex to capture realistic problem characteristics such as routing complexity, capacity constraints, and spatial diversity.

Table 8: Relative performance gap (%) of alternative sub-GCP configurations compared to our aligned configuration on LRP instances.

| Sub-GCP Configuration | 100 customers | 200 customers | Average |
|---|---|---|---|
| Simplified Sub-GCPs | -6.62 | -7.89 | -7.26 |
| Partial Sub-GCPs | -2.68 | -2.94 | -2.81 |
| Misaligned Sub-GCPs | -7.96 | -9.37 | -8.67 |

Negative values indicate performance degradation relative to the aligned configuration.

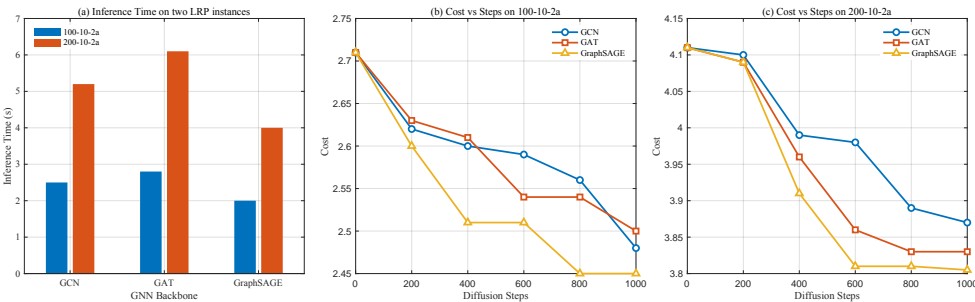

Figure 4: Performance comparison of three GNN backbones in our RLG-DM on two typical LRP instances

These results reveal that effective sub-GCP selection requires satisfying two key properties: structural alignment between sub-GCPs and the target problem, and gradual complexity progression through intermediate stages. Performance degradation scales consistently with problem size across all configurations, supporting the principle that hierarchical decomposition becomes more valuable as complexity grows. The sensitivity patterns are systematic rather than arbitrary, with structural misalignment causing more severe degradation than insufficient complexity, which in turn exceeds the impact of skipping intermediate stages. Notably, the misaligned configuration demonstrates that transfer learning cannot overcome fundamental structural mismatch, while the partial configuration shows that some benefits persist even without fine-grained progression.

Based on these findings, we recommend three practical guidelines for sub-GCP selection: (1) choose sub-GCPs that share core optimization patterns with the target problem (e.g., spatial sub-GCPs for spatial problems, constraint-satisfaction sub-GCPs for scheduling problems); (2) use sub-GCP instances with realistic problem sizes (e.g., 50-100 nodes) rather than toy examples; and (3) include intermediate complexity levels to facilitate smooth knowledge transfer.

## A.4 GNN BACKBONE COMPARISON

We also evaluate the performance of three GNN backbones (GCN Jiang et al. (2019), GAT Veličković et al. (2017), and GraphSAGE Hamilton et al. (2017)) on two typical LRP instances (100-10-2a and 200-10-2a), as shown in Fig.4. In Fig.4(a), all architectures show an increased inference time as the number of customers grows from 100 to 200. However, GraphSAGE maintains significantly lower inference time, demonstrating its suitability for large-scale LRP tasks. Figs.4(b) and (c) show the cost reduction over diffusion steps. GraphSAGE converges faster and reaches the lowest objective value with fewer steps than both GCN and GAT. These results indicate that GraphSAGE is the most efficient GNN backbone for solving LRP in our framework, offering the best balance between inference time and optimization performance. Therefore, GraphSAGE is employed as the GNN backbone in RLG-DM.

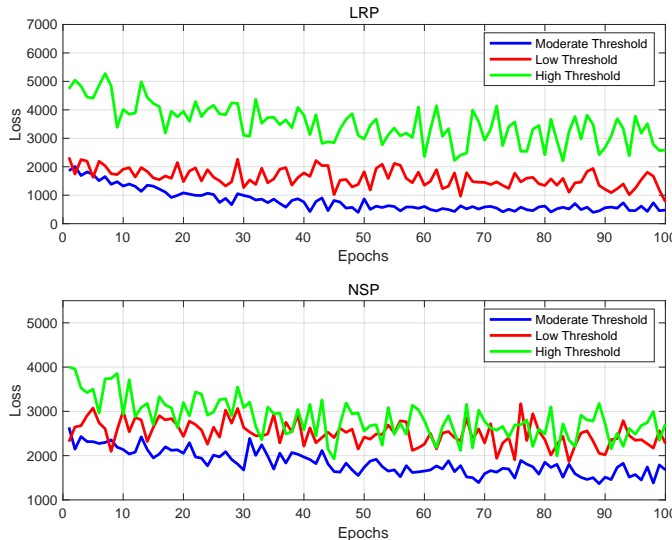

Figure 5: Convergence curves for different threshold settings in the LRP and NSP tasks

### A.5 PARAMETER SENSITIVITY AND TRAINING DYNAMICS

#### A.5.1 ANALYSIS OF THRESHOLDS IN ADAPTIVE NOISE SCHEDULING

We evaluate the impact of varying threshold settings for adaptive noise scheduling on the performance of the RLG-DM across both the LRP and NSP problems. We tested three threshold settings: moderate, low, and high. Under the moderate threshold, the cosine schedule is applied when $0.05 < |IG(X_t)| \leq 0.1$, and the exponential schedule is used when $|IG(X_t)| > 0.1$; under the low threshold, the cosine schedule is applied when $0.02 < |IG(X_t)| \leq 0.05$, and the exponential schedule is used when $|IG(X_t)| > 0.1$; under the high threshold, the cosine schedule is applied when $0.1 < |IG(X_t)| \leq 0.15$, and the exponential schedule is used when $|IG(X_t)| > 0.15$.

Figure 5 shows the convergence curves for different threshold settings based on the experimental results for the LRP and NSP tasks. For both LRP and NSP, the moderate threshold provided the best trade-off between convergence speed and training efficiency. It also showed the fastest convergence and the lowest final loss, while maintaining flexibility throughout the training process. In contrast, the high threshold resulted in slower convergence and higher final loss, whereas the low threshold offered more stability but was slower than the moderate threshold.

#### A.5.2 DYNAMICS OF LAYER IMPORTANCE WEIGHTS

We further examine the evolution of the learnable layer importance weights $\varsigma_l$ to understand how the model prioritizes different hierarchy levels. Figure 6 depicts the training dynamics on the LRP task. Starting from a uniform initialization (0.5, 0.5), the weights converge to approximately (0.62, 0.38) after 50k steps. This indicates that Level 1 (FLP) decisions exert a stronger influence on the global objective than Level 2 (VRP) routing details. This automatic adjustment mechanism effectively captures the hierarchical dependency of the problem, eliminating the need for manual weight tuning.

### A.6 COMPUTATIONAL EFFICIENCY ANALYSIS

We evaluate the computational efficiency of RLG-DM in comparison to baseline methods to address concerns regarding the potential overhead of the hierarchical control loop. While our framework involves a hierarchy of levels and subproblems, the inference process is computationally efficient due to parallelization. Subproblems within each level are vectorized and executed as a single batched graph operation. The overall complexity is $O(T \times L)$, where $T$ is the number of diffusion steps and $L$ is the number of hierarchical levels. Since $L$ is typically small (2-3 for most multilevel GCPs),

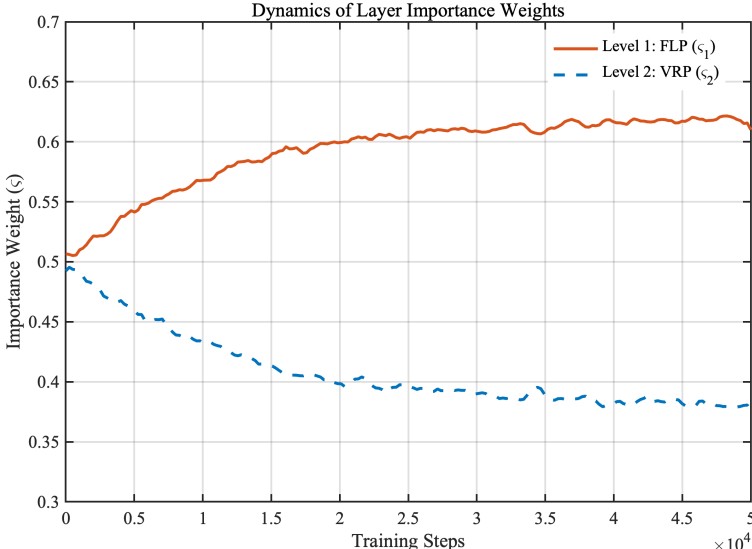

Figure 6: Evolution of learnable layer importance weights $\varsigma_l$ during training on the LRP task. Starting from a uniform initialization (0.5, 0.5), the weights converge to approximately (0.62, 0.38). This divergence demonstrates that the model automatically learns the hierarchical synergy, prioritizing strategic Facility Location decisions (Level 1) over local Routing refinements (Level 2) to optimize the global objective.

the complexity scales near-linearly with diffusion steps. Furthermore, our GNN backbone (Graph-SAGE) operates with $O(N+E)$ complexity per layer, which is more scalable than the $O(N^2)$ attention mechanisms used in some transformer-based baselines. A key distinction lies in our end-to-end generation approach. Baselines such as DIFUSCO and T2T rely on post-hoc decoding procedures (e.g., MCTS or sampling for DIFUSCO; gradient-based search for T2T) to obtain valid solutions from learned distributions. In contrast, RLG-DM directly generates valid graph structures through RL-guided denoising, avoiding the overhead of iterative refinement procedures.

As illustrated in Figure 7, we compare the inference latency on LRP instances. For 100-customer instances, RLG-DM requires 1.8s, whereas DIFUSCO and T2T require 8.5s and 7.9s, respectively. As the problem scale increases to 200 customers, RLG-DM maintains efficient inference at 4.2s, compared to DIFUSCO (21.5s) and T2T (19.8s). This represents an improvement of approximately 5× in inference speed, validating that our hierarchical controller effectively balances solution quality with computational efficiency.

## A.7 RELATED WORK

Recent studies have demonstrated the potential of DMs in addressing GCP Yao et al. (2024); Wang et al. (2025); Zhang et al. (2024), where the solution space is inherently discrete and structurally complex. Unlike conventional generative tasks, combinatorial optimization requires guided sampling toward high-quality, constraint-satisfying solutions. To this end, several work Sun & Yang (2023); Sanokowski et al. (2024b) have adapted diffusion processes to discrete domains by embedding combinatorial structures (e.g., permutations, graphs) into continuous latent spaces suitable for diffusion-based generation. In routing and scheduling tasks, for instance, DMs are trained to denoise perturbed versions of solution graphs or sequences, thereby learning distributions over feasible and near-optimal solutions Joshi et al. (2022). Nevertheless, existing methods Sanokowski et al. (2024b); Lin et al. (2024) have primarily focused on flat combinatorial problems and remain limited in addressing hierarchical dependencies across multiple graph layers. Unlike prior work focusing on flat structures, we introduce a reinforcement-guided diffusion framework that integrates structural priors and value-based guidance to solve multilevel GCPs.

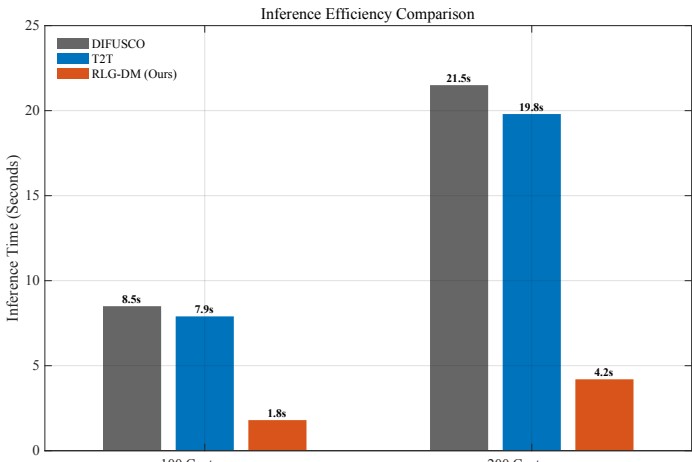

Figure 7: Inference efficiency comparison. RLG-DM achieves approximately $5\times$ speedup compared to diffusion baselines (DIFUSCO and T2T) by generating valid solutions end-to-end without iterative refinement procedures.

### A.8 MULTILEVEL GCPS FORMULATION

The optimization problem for multilevel GCPs can be expressed as follows, where the total performance across all layers must be maximized (or minimized), while ensuring the consistency and feasibility of decisions across layers:

$$
\begin{aligned}
\text{Max (or Min)} \quad & \sum_{k=1}^{K} f_k(V_k', E_k') \\
\text{s.t.} \quad & \sum_{k=1}^{K} C_k(V_k', E_k') \\
& \sum_{k=1}^{K} g_k(V_k', E_k') \le \delta, \quad \forall k \\
& \sum_{k=1}^{K} h_k(V_k', E_k') = \eta, \quad \forall k \\
& \Phi(V_k', V_{k+1}') \le \gamma, \quad \forall k \text{ and } k+1
\end{aligned}
\tag{16}
$$

Where:

- $f_k(V_k', E_k')$ is the objective function for the subgraph $G_k'$ at layer $k$,
- $C_k(V_k', E_k')$ are the constraints specific to each layer $k$,
- $g_k(V_k', E_k') \le \delta$ and $h_k(V_k', E_k') = \eta$ impose constraints on layer $k$,
- $\Phi(V_k', V_{k+1}') \le \gamma$ represents the coupling constraints between adjacent layers, ensuring that decisions made in one layer are consistent with those made in other layers.

The key feature of this model is the inclusion of coupling constraints $\Phi(V_k', V_{k+1}')$, which are essential for capturing the hierarchical dependencies between the layers. These constraints ensure that the optimization of one layer does not violate the constraints or optimality conditions of other layers. The interaction between layers is governed by these constraints, making the problem significantly more complex than single-level GCPs. Each layer's optimization process is influenced by the decisions made at other levels, and the feasibility of a solution is determined by the consistency of these decisions across the hierarchy. The coupling constraints $\Phi$ act as a bridge between the layers, ensuring that the solution respects the interdependencies and that the overall system behaves as a unified whole.

Moreover, we summarize the key mathematical notations and symbols used throughout this paper in Table 9.

Table 9: Key notation used throughout the paper

| Notation | Description |
| --- | --- |
| $G = (V, E)$ | Graph with node set $V$ and edge set $E$ |
| $X_t$ | Graph state at diffusion step $t$ |
| $\mathbf{h}_i^t$ | Embedding of node $i$ at step $t$ |
| $\mathbf{A}_{l,i}^t$ | Adjacency matrix for subproblem $i$ at level $l$ |
| $s_t$ | MDP state (concatenated embeddings) |
| $a_t$ | Action selected at step $t$ |
| $Q_{l,i}(s, a)$ | Q-function for subproblem $i$ at level $l$ |
| $\varsigma_l$ | Learnable importance weight for level $l$ |
| $\lambda$ | Penalty weight for constraint violations |
| $\gamma$ | Discount factor |
| $T$ | Number of diffusion steps |
| $L$ | Number of hierarchical levels |
| $M_l$ | Number of subproblems at level $l$ |

## A.9 IMPLEMENTATION DETAILS

### A.9.1 ACTION SPACE SPECIFICATIONS

To provide a concrete example of our MDP formulation, we detail the action space specification for the LRP. The decision process operates across two levels. At Level 1 (FLP), the agent manages facility configurations using two discrete actions: $a_{\text{open}}(i)$ to open facility $i$, and $a_{\text{close}}(i)$ to close it. At Level 2 (VRP), the agent optimizes routes for the open facilities. The action space includes $a_{\text{insert}}(c, k, p)$ to insert a customer $c$ into route $k$ at position $p$, $a_{\text{swap}}(c_1, c_2)$ to exchange the positions of two customers, and $a_{\text{2-opt}}(e_1, e_2)$ to perform standard 2-opt edge exchanges. Executing any of these actions directly modifies the corresponding adjacency matrix, thereby inducing a deterministic state transition in the MDP.

Similarly, for NSP, Level 1 (GC) actions assign shift constraints to nurses, while Level 2 (AP) actions swap specific shifts to balance workload. For FJSP, Level 1 (AP) allocates operations to machines, Level 2 (TSP) determines the processing sequence on each machine, and Level 3 (GC) resolves temporal conflicts. All actions are implemented as discrete modifications to the adjacency matrix or node features.

### A.9.2 DQN IMPLEMENTATION DETAILS

Each subproblem Q-function $Q_{l,i}(s, a)$ is implemented as a multi-layer perceptron (MLP) featuring two hidden layers with dimensions [512, 256] and ReLU activation functions. Note that this is separate from the GNN encoder, which uses a hidden dimension of 64 as specified in Section 4.1. This architecture processes the concatenated node embeddings to output value estimates for all available actions. The model is trained using an experience replay buffer with a capacity of 10,000 transitions and a mini-batch size of 64. To ensure stability, we employ a target network that is updated periodically. Optimization is performed using the Adam optimizer with a learning rate of 0.001. For exploration, we utilize an $\varepsilon$-greedy strategy where $\varepsilon$ is linearly annealed from 1.0 to 0.1 over the training course. The discount factor is set to $\gamma = 0.99$ to encourage long-term planning. The penalty weight $\lambda$ is selected from the range $[3.0, 10.0]$ based on the specific problem instance, calibrated such that constraint violations account for approximately 10–20% of the total reward magnitude. Finally, the layer importance weights $\varsigma_l$ are initialized uniformly and updated end-to-end via gradient descent based on the Q-learning objective.

## A.10 PSEUDOCODE FOR THE FORWARD DIFFUSION, REVERSE DIFFUSION, AND INFERENCE PHASES

In our RLG-DM framework, Algorithm 1 presents the pseudocode for learning structural priors through forward diffusion. Algorithm 2 outlines the reverse diffusion training process guided by multilevel Q-learning. Finally, Algorithm 3 describes the inference procedure for optimal solution generation via reinforcement-guided reverse diffusion.

---

**Algorithm 1** Learning structural priors via forward diffusion

---

**Require:** Training set $\mathcal{D}_{\text{simple}}$ with optimal solutions; diffusion steps $T$
**Ensure:** Pretrained GNN encoder $\text{GNN}_\theta$
1: **for** each $(G, X_0)$ in $\mathcal{D}_{\text{simple}}$ **do**
2:      Initialize graph state $X_0$            ▷ Graph structure and node features
3:      Compute initial embedding $\mathbf{h}_0 \leftarrow \text{GNN}_\theta(X_0)$            ▷ $\mathbf{h}_0 \in \mathbb{R}^{n \times d}$
4:      Initialize total loss $\mathcal{L} \leftarrow 0$
5:      Initialize noise schedule to linear
6:      **for** $t = 1$ to $T$ **do**
7:          Compute noise level $\sigma_t$ with adaptive scheduling
8:          Sample noise $\epsilon_t \sim \mathcal{N}(0, \sigma_t^2 \mathbf{I})$            ▷ Noise for embeddings
9:          Encode graph with noisy aggregation $\mathbf{h}_t \leftarrow \text{GNN}_\theta(X_0, \epsilon_t)$        ▷ Via Eq. 5
10:         Compute loss $\mathcal{L}_t$ as in Eq. equation 7
11:         Accumulate $\mathcal{L} \leftarrow \mathcal{L} + \mathcal{L}_t$
12:      **end for**
13:      Update $\text{GNN}_\theta$ using Adam to minimize $\mathcal{L}$
14: **end for**

---

**Algorithm 2** Reverse Diffusion Training via Multilevel Q-Learning

---

**Require:** Training set $\mathcal{D}_{\text{train}}$ with solved GCPs, pretrained GNNs $\{\text{GNN}_l\}$, total steps $T$
**Ensure:** Trained Q-functions $\{Q_{l,i}\}$ **and layer weights** $\{\varsigma_l\}$
1: **for** each $(G, X_0)$ in $\mathcal{D}_{\text{train}}$ **do**
2:      Inject noise to obtain initial $X_T$
3:      **for** $t = T$ to $1$ **do**
4:          Encode state $s_t$ via frozen GNNs
5:          Select action $a_t$ using $\varepsilon$-greedy from $Q_{\text{joint}}(s_t, a)$
6:          Apply $a_t$ to obtain updated graph $X_{t-1}$
7:          Observe reward $r_t$
8:          **for** $l = 1$ to $L$, $i = 1$ to $M_l$ **do**
9:             Compute TD target: $y_t = r_t + \gamma \max_{a'} Q_{l,i}(s_{t+1}, a')$
10:            Update Q-function: $Q_{l,i}(s_t, a_t) \leftarrow Q_{l,i}(s_t, a_t) + \eta\Big(y_t - Q_{l,i}(s_t, a_t)\Big)$
11:          **end for**
12:          Update importance weights $\varsigma_l$ via gradient descent on joint loss
13:      **end for**
14: **end for**

---

---

**Algorithm 3** Inference for optimal solution generation via reinforcement-guided reverse diffusion

---

**Require:** Multilevel GCP instance $G$ with $L$ layers, pretrained GNNs $\{\text{GNN}_l\}$, trained Q-functions $\{Q_{l,i}\}$, subproblem counts $\{M_1, \ldots, M_L\}$, diffusion steps $T$
**Ensure:** Final optimized solution $X_0$
 1: Initialize fully noise-injected graph $X_T$
 2: **for** $t = T$ to 1 **do**
 3:     **for** $l = 1$ to $L$ **do**
 4:         **for** $i = 1$ to $M_l$ **do**                                      $\triangleright$ Executed in parallel
 5:             Compute embeddings: $\mathbf{H}_{l,i}^t \leftarrow \text{GNN}_l(\mathbf{A}_{l,i}^t, \mathbf{H}_{l-1}^t)$
 6:         **end for**
 7:     **end for**
 8:     Construct state: $s_t \leftarrow [\mathbf{H}_{1,1}^t, \ldots, \mathbf{H}_{L,M_L}^t]$
 9:     **for** $l = 1$ to $L$ **do**
10:         Compute $Q_l(s_t, a)$ using Eq. (10)
11:     **end for**
12:     Compute $Q_{\text{joint}}(s_t, a)$ using Eq. (11)
13:     Select action: $a_t \leftarrow \arg\max_a Q_{\text{joint}}(s_t, a)$
14:     Apply $a_t$ to obtain $X_{t-1}$
15: **end for**
16: **return** $X_0$ after applying repair operator $\mathcal{R}(\cdot)$

---

## A.11 Limitations and Future Directions

The proposed approach relies on domain-informed problem decomposition, utilizing established multilevel structures from the literature (e.g., LRP decomposes into FLP and VRP; NSP into GC and AP). This requires identifying relevant sub-GCPs and pretraining their encoders before tackling the target multilevel problem. However, this modular design offers significant advantages. Once pretrained, encoders generalize across problem instances and scales. For example, a single FLP encoder trained on 20-50 facilities can handle LRP instances with 5-100 facilities. Practitioners can build a library of pretrained encoders and compose them for new multilevel problems as needed. Our results on FJSP and multi-echelon LRP demonstrate effective scalability to larger problems and deeper hierarchies without retraining. The primary remaining challenge is automatically discovering optimal decompositions for novel problem structures where domain knowledge is sparse. Future work will explore meta-learning approaches to identify compositional patterns from data, enabling broader automation of the framework.

