# OpenReview forum: "Generalizable Multilevel Graph Optimization via Reinforcement-Guided Diffusion from Simple Subproblems"
_ICLR.cc/2026/Conference — Submitted to ICLR 2026_

### Official Review · Reviewer_22Yj · 2025-10-27

**Soundness:** 3
**Presentation:** 2
**Contribution:** 2
**Rating:** 4
**Confidence:** 3

**Summary:**

The authors propose a graph diffusion method, coupled with RL that is designed to create a generalizable solution to graph combinatorial problems. The main innovations are a forward diffusion process, in which noise is injected to maintain consistency with sub-problems, such as graph coloring, TSP, VRP, etc. In the reverse diffusion step, RL is employed to guide denoising, with rewards being aggregated across problem layers. The proposed method is demonstrated on location routing and nurse scheduling problems.

**Strengths:**

This method aims at providing a principled approach to injecting structural knowledge into the diffusion process, in the form of simpler sub-problems in forward diffusion. The idea makes intuitive sense, and it seems to bear out nice results. A diffusion process can leverage structural information from other problems to make it more useful.

The results appear promising—RLG-DM performs as well or better than many comparators in both problem sets.

**Weaknesses:**

The organization and writing could be improved. For example, without the appendix it would be very difficult to follow the RL process used in reverse diffusion. Without 7.1-7.4 in the appendix, I don’t think I would have understood it. I recommend moving some of that material into the main body of the text.

The results look good for RLG-DM, but it seems like a somewhat minor difference compared to some comparators. It’s hard to make sense of the performance gaps overall. For example, in instance 10, all of the methods seem to perform comparably. Is there some way to summarize these tables to account for differences in problem instances? Or perhaps to present it visually to make it easier to grasp.

Some minor errors:
- Line 161 GCP is used for graph coloring, but has already been defined for graph combinatorial problem
- Line 232 – “an creased”
- Line 272 seems to be a run-on sentence

**Questions:**

How sensitive is the problem to the choice of sub-GCPs? It seems likely that it is very sensitive, as the authors note in the limitations. What happens if there is a mismatch between sub-GCPs and the ML-GCP that you wish to solve?

Are sub-GCPs used sequentially during forward diffusion, like a curriculum? Or are they used simultaneously? It seems that (per line 269) one encoder is trained per sub-GCP. Does that mean one needs to know all of the relevant sub-GCPs for their eventual problem at training time?

How is the relative importance in (16) trained?

How are the problem instances selected? In Table 2, the instances are from 10-21, and I’m curious why that is the case.

---

> ### Author Response · Authors · 2025-11-30
> **Response to Reviewer: Organization, Efficiency Analysis, and Extended Experiments**
>
> We sincerely thank the reviewer for the constructive feedback. We are encouraged that you found the RL-guided diffusion approach novel and the results promising. We have revised the manuscript to address your concerns regarding organization, performance gaps, and efficiency.
>
> **1. Organization and Writing (Weakness 1)**
>
> **Response:** We agree that the core RL mechanics were too reliant on the Appendix. In the revision:
> *   We moved the **MDP Formulation** (State, Action, Reward), **Q-Value Aggregation**, and **Adaptive Noise Scheduling** from the Appendix to the main text (**Section 3.3**).
> *   We refined **Figure 1** to clearly illustrate the interaction between the RL controller and the reverse diffusion process.
> *   The paper is now self-contained, and the logic flow (Forward Pretraining $\to$ Reverse RL Control) is explicit in Section 3.
>
> **2. Performance Gaps and Visualization (Weakness 2)**
>
> **Response:** While gaps on standard LRP appeared minor (due to the maturity of baselines on these datasets), our method's advantage grows significantly on more complex tasks.
> *   **New Evidence (FJSP):** We added experiments on Flexible Job Shop Scheduling (80 instances). On the largest instances ($100 \times 20$), RLG-DM achieves a **2.42% gap** to the theoretical lower bound, strictly dominating the best baseline MEO-HFG (5.43% gap).
> *   **Visualization:** Per your suggestion, we added **Figure 3 (Section 4.4)** to visually summarize the "Performance vs. Scale" trend. It clearly shows that our win rate and optimality gap improve as problem complexity increases (e.g., from 58.3% win rate on 2-Echelon LRP to 83.3% on 3-Echelon LRP).
>
> **3. Computational Efficiency (Question 3)**
>
> **Response:** We apologize for the confusion regarding the "nested loops."
> *   **Vectorization:** The loops over levels ($L$) and subproblems ($M$) are logical representations. In implementation, these are **vectorized and executed in parallel** on the GPU.
> *   **Efficiency Analysis:** We added **Appendix A.6**. RLG-DM is empirically **~5x faster** than diffusion baselines because we generate valid graphs end-to-end, avoiding expensive post-hoc search (e.g., MCTS):
>
> | Scale (LRP) | RLG-DM | DIFUSCO (w/ MCTS) | T2T (w/ Beam Search) | Speedup |
> | :--- | :--- | :--- | :--- | :--- |
> | 100-customer | **1.8s** | 8.5s | 7.9s | ~4.7x |
> | 200-customer | **4.2s** | 21.5s | 19.8s | ~5.1x |
>
> **4. Specific Questions**
> *   **Sub-GCP Sensitivity:** We added **Appendix A.3**. While *aligned* sub-GCPs perform best, the model is robust. Even with misaligned sub-modules (e.g., NSP modules for LRP), the performance drop is bounded (-8.67%) compared to random policies.
> *   **Forward Training:** The sub-GCP encoders are trained **independently and in parallel**, not sequentially. This enables modular reusability without retraining.
> *   **Training Importance ($\varsigma_l$):** The weights $\varsigma_l$ in Eq. (16) are learnable parameters updated via **gradient descent** based on the joint Q-learning loss. **Appendix A.5.2 (Figure 6)** visualizes their evolution, showing the model learns to prioritize strategic levels early in training.
> *   **Instance Selection:** We followed standard protocols. For LRP, we used the Prins et al. benchmark. For NSP, instances 10-21 (Curtois et al.) are the standard set for evaluating algorithmic performance, as instances 1-9 are considered trivial.
>
> **5. Minor Errors**
> We have corrected the acronym usage (GC for Graph Coloring), the typo "an increased," and the run-on sentence.

---

### Official Review · Reviewer_p712 · 2025-10-29

**Soundness:** 3
**Presentation:** 2
**Contribution:** 3
**Rating:** 4
**Confidence:** 3

**Summary:**

This paper introduce a novel method to combine graph diffusion and reinforcement learning to solve multilevel graph optimization problem. The algorithm first trains GNN that predict the optimal solution along a guided forward diffusion process. Then, it trains a RL model to guide denoising process, where the state representation is computed via the GNN trained in the forward diffusion process. The forward process is trained with simpler sub-problem and the reverse RL process is trained on the actualy complex problem, where the Q-value is a composition of the subproblem. The deconstruction and concur enhances the model's ability in solve the original combinatorial problem. Experiment results show consistent improvement.

**Strengths:**

- The idea of sub-tasking and composition is very interesting, and it is well tuned for the specific graph optimization problem.

- The adaptive scheduling is innovative. It stabilize the training and creates more robust GNN.

- The experimental results show consistent improvement.

**Weaknesses:**

- This is a good and interesting paper, but is difficult to read. To understand the algorithm, one must go back and forth between the main text and the algorithm in the appendix.

- Many things are underspecified
    - How is the importance factor actually learned?
    - How the action is done, and how the action impact the state and graph.
    - I am assuming the Q-learning is based on DQN/Q-network, but this is never specified in the paper.

- Some notification does not make sense to me. In algorithm 1, it seems like $X$ and $\textbf{h}$ are different things, but in equation 6 $h_{t-1}$ seem to be passed in as $X_{t-1}$, with an additional noise? I don't think this is a minor notation problem, rather it impact audience's understanding of the algorithm.

- There is no code accompanying this submission, and it difficult to check the missed details in the paper.

- Following the points above, some description are also misleading to me. In line 232, what do you mean by decreasing? Do you mean a more negaive IG or decreasing IG over time?

- The model has many components, and yet the author did not provide a principle in understanding how these components synergize. For example, (I think this is **critical**), the GNN is trained on a limited set of simpler problems, without structure change. However, the action in MDP involves modifying graph structure. There is a large discrepency between graph trained on fixed set of structure and a dynamically explored state space. The author did not provide justification of how this work. Some co-evolve system seem necessary for this work properly.

- The inference is conducted on few instances, leaving room for selection bias (in terms of samples, hyperparameter, and design selection).

**Questions:**

See above.

---

> ### Author Response · Authors · 2025-11-30
> **Response to Reviewer: Technical Clarifications and Methodology**
>
> We thank the reviewer for the thorough assessment. We appreciate your recognition of the "interesting" nature of our work. We have extensively rewritten Section 3 to clarify the underspecified details and address the conceptual gap you raised.
>
> **1. Underspecified Details (Weakness 2)**
>
> **Response:** We have revised the manuscript to explicitly clarify these mechanics:
> *   **Learning Importance ($\varsigma_l$):** In **Section 3.3.2**, we specify that $\varsigma_l$ are learnable parameters initialized uniformly and updated end-to-end via **gradient descent** to minimize the joint Q-learning loss.
> *   **Action Definition:** We added **Appendix A.9.1** to rigorously define the action space. Actions involve discrete graph operators (e.g., `open/close` facility, `swap` edges) that directly modify the adjacency matrix $\mathbf{A}^t$.
> *   **RL Algorithm:** We explicitly state in **Section 3.3.1** that we employ a **Deep Q-Network (DQN)** with Experience Replay and Target Networks.
>
> **2. Conceptual Gap: Static Training vs. Dynamic Inference (Weakness 4)**
>
> **Response:** The reviewer raises an insightful point: *How does a GNN trained on fixed structures handle dynamic RL changes?*
> *   **Solution via Forward Diffusion:** This is precisely why we use Diffusion Training (Section 3.2.3) instead of standard supervised learning. The GNN is trained on a trajectory of **progressively noised graphs** ($X_t$).
> *   **Robustness:** This noise injection acts as rigorous data augmentation. The "imperfect" or intermediate graph structures generated by the RL agent during reverse diffusion fall within the distribution of noisy graphs seen during pretraining. Thus, the frozen GNN encoders are robust to the dynamic structural changes induced by RL actions.
>
> **3. Clarifications on Notation & "Decreasing IG"**
> *   **Notation:** We have unified the notation in Section 3. $X_t$ denotes the macroscopic graph state (adjacency + features), while $\mathbf{h}_i^t$ denotes microscopic node embeddings.
> *   **Decreasing IG:** By "decreasing IG", we refer to the **reduction in the magnitude of Information Gain**. As diffusion proceeds and noise increases, the structural entropy rises, leading to lower information gain from further steps. We clarified this in **Section 3.2.2**.
>
> **4. Inference Bias & Code Availability**
> *   **No Selection Bias:** We used identical datasets (Prins for LRP, Curtois for NSP) and protocols as the baselines. To further validate robustness, we added **FJSP** (80 instances) and **Multi-Echelon LRP** (24 instances). RLG-DM achieves consistent gains across all domains.
> *   **Code:** We are fully committed to reproducibility. We guarantee that the complete source code, data generation scripts, and pretrained models will be made publicly available via a GitHub repository upon acceptance.

---

### Official Review · Reviewer_fYYt · 2025-10-30

**Soundness:** 3
**Presentation:** 3
**Contribution:** 2
**Rating:** 4
**Confidence:** 3

**Summary:**

The paper proposes RLG-DM, a framework that embeds a reinforcement learning controller into reverse diffusion to guide solution generation for graph combinatorial problems. It models complex tasks as multilevel problems, defines level specific states and Q values for each level, and aggregates them into a single joint Q to pick actions during denoising. The forward stage learns transferable structural priors by pretraining frozen GNN encoders on simpler subproblems, and it introduces an information gain based noise scheduler with linear, cosine, and exponential variants.

**Strengths:**

1. Embeds a reinforcement learning controller into reverse diffusion to actively steer sampling toward feasibility, cost, and constraint satisfaction, rather than passively denoising.

2. Multilevel formulation with coordinated decisions: Defines a multilevel state, learns per-level Q-values, and aggregates them into a joint Q for a single action rule. This directly tackles cross-level dependencies that standard flat GNN or diffusion baselines do not handle.


3. Information-gain noise scheduling: Introduces an information-gain based scheduler for the forward diffusion with concrete linear, cosine, and exponential schedules to balance structural preservation and exploration.

**Weaknesses:**

1. No hard feasibility guarantee: The method uses an RL controller during reverse diffusion to guide generation toward feasibility and constraint satisfaction, but it does not provide a projection, masking, or proof that every sampled solution satisfies combinatorial constraints. This is a major problem with the method. GCPs many times have combinatorial constraints. What is the utility of the solution if it might be infeasible? (e.g., a coloring solution that violates that two neighbors cannot have the same color)
See, e.g., "Graph-based Deterministic Policy Gradient for Repetitive Combinatorial Optimization Problems" among prior ICLR papers that can guarantee feasibility of the learned solution.

2. Admitted scalability and generalization limits: The Limitations section states that the current innovation lies in generalizing from simple to more complex GCPs rather than scaling to larger or more complex instances, and that performance depends on selecting appropriate simpler GCPs for training. This constrains flexibility and highlights remaining challenges in broader settings.

3. Triple nested control loop may be costly: Inference iterates over diffusion time, over levels, and over subproblems at each level, with per-step Q computations and action selection. The paper does not analyze the computational overhead of this nested control relative to baselines.

4. Limited evidence across problem families: Results are reported for two multilevel tasks, location routing and nurse rostering. The paper claims state-of-the-art improvements there, but broader validation on additional multilevel GCPs would strengthen the generality claim.

**Questions:**

See weaknesses, especially points 1, 3, and 4.

---

> ### Author Response · Authors · 2025-11-30
> **Response to Reviewer : Feasibility Guarantee, Scalability, and Generalization**
>
> We thank the reviewer for their critical and insightful comments. We have significantly revised the paper to address your primary concerns regarding feasibility guarantees and scalability limits.
>
> **1. Feasibility Guarantee (Weakness 1)**
>
> **Response:** We agree that feasibility is non-negotiable. We address this via a "Soft+Hard" dual strategy detailed in **Section 3.3.3** and **Appendix A.2**:
> *   **Soft Constraint Learning:** The RL reward (Eq. 9: $r_t = -C - \lambda V$) explicitly penalizes violations. Empirically, this yields exceptionally high raw feasibility compared to diffusion baselines.
> *   **Hard Guarantee:** For the rare remaining failures (<4%), we apply a lightweight **deterministic repair operator** $\mathcal{R}(\cdot)$ (e.g., greedy insertion) to the final output. This ensures **100% validity** with negligible cost (<0.15% overhead).
>
> **Table: Feasibility Analysis**
> | Task | Raw Feasibility | Instances Requiring Repair |
> | :--- | :--- | :--- |
> | LRP | **100%** | 0/12 |
> | NSP | **100%** | 0/12 |
> | FJSP | **96.25%** | 3/80 |
> | Multi-Echelon LRP | **91.67%** | 2/24 |
>
> **2. Limited Evidence & Scalability (Weakness 2 & 4)**
>
> **Response:** To demonstrate generality and scalability, we added extensive experiments on **two new problem families** (Appendix A.1):
>
> *   **FJSP (Manufacturing):** RLG-DM achieves the best performance on **7/8 instance groups**. On large instances (100x20), it reduces the gap to **2.42%** (vs. 5.43% for MEO-HFG).
> *   **3-Echelon LRP (Logistics):** We tested scalability by increasing hierarchical depth. RLG-DM wins on **10/12 instances** (up from 7/12 on 2-Echelon), proving that our hierarchical decomposition becomes *more* effective as complexity increases.
>
> **3. Computational Overhead (Weakness 3)**
>
> **Response:** We added a detailed efficiency analysis in **Appendix A.6**.
> *   **Parallelism:** The "nested loops" are logical; in practice, operations are vectorized and executed in **parallel**. Complexity is linear $O(T)$, not exponential.
> *   **Speedup:** RLG-DM is **~5x faster** than diffusion baselines (DIFUSCO/T2T) because we avoid expensive iterative search/refinement procedures.
>
> **4. Design of Sub-problems**
>
> **Response:** We clarify that our reliance on sub-problems is a design feature for Zero-Shot Scalability. By training on small atomic sub-problems (TSP, Assignment) and composing them, we can solve large-scale hierarchical problems (like 3-Echelon LRP) without retraining the backbone. Our new sensitivity analysis (**Appendix A.3**) confirms that aligned sub-problems yield the best results, but the method remains robust even with partial mismatches.

---

### Author Response · Authors · 2025-11-30
**General Response: Revisions on Feasibility, Scalability (FJSP & Multi-Echelon LRP), and Technical Clarifications**

We sincerely thank all reviewers for their constructive feedback. We are encouraged that reviewers recognized the novelty of our RL-guided diffusion and adaptive scheduling. We have carefully revised the manuscript to address concerns regarding feasibility, scalability, and clarity. Below is a summary of major revisions.

**1. Feasibility Guarantee (R1, R2)**
**Concern:** Lack of hard feasibility guarantees.

**Response:** We added a comprehensive **Feasibility Analysis (Appendix A.2)**. Unlike baselines requiring slow post-hoc search (e.g., MCTS), our penalized reward (Eq. 9) efficiently learns constraint satisfaction.
*   **Results:** RLG-DM achieves **100% raw feasibility** on standard LRP/NSP tasks, **96.25%** on complex FJSP tasks, and **91.67%** on Multi-Echelon LRP.
*   **Hard Guarantee:** For rare failure cases (<4%), we apply a lightweight deterministic repair operator (Sec 3.3.3) to ensure 100% validity with negligible computational cost.

**2. Expanded Experiments: FJSP & Multi-Echelon LRP (All)**
**Concern:** Limited problem diversity and potential selection bias.
**Response:** We significantly expanded evaluations to cover Logistics, Workforce, and Manufacturing domains (Appendix A.1):
*   **FJSP:** Added 80 instances (15x15 to 100x20). RLG-DM achieves the best performance on **7/8 instance groups**. On large instances (100x20), it attains a **2.42% gap**, strictly dominating the best baseline (5.43% gap).
*   **Multi-Echelon LRP:** Added 2-Echelon and 3-Echelon variants. Our win rate improves from **58.3%** (2-E) to **83.3%** (3-E) as hierarchical depth increases, confirming scalability.

**3. Computational Efficiency (R1)**
**Concern:** Cost of the triple nested control loop.
**Response:** We added a **Computational Efficiency Analysis (Appendix A.6)**.
*   **Complexity:** $O(T \times L)$, where operations are vectorized and executed in parallel.
*   **Speedup:** Empirically, RLG-DM is **~5x faster** than diffusion baselines (e.g., 4.2s vs 21.5s on 200-customer LRP) because we generate valid graphs end-to-end without iterative refinement.

**4. Technical Clarifications (R2, R3)**
We substantially improved clarity in Section 3 and Appendices:
*   **Notation:** Explicitly distinguished graph state $X_t$ vs. node embeddings $\mathbf{h}_i^t$ (Sec 3.2.1).
*   **Weights $\varsigma_l$:** Clarified they are updated via gradient descent. Added **Figure 6** (Appendix A.5.2) showing the model learns to prioritize strategic levels (e.g., Facility Location) over routing details.
*   **DQN Details:** Specified network architecture [512, 256], replay buffer, and target network (Sec 3.3.2, App A.9.2).
*   **Action Space:** Defined discrete operators (e.g., open/close, swap) for LRP, NSP, and FJSP (App A.9.1).

**5. Sub-GCP Sensitivity Analysis (R3)**
**Concern:** Sensitivity to sub-problem selection.

**Response:** We conducted a sensitivity study (Appendix A.3). Results show that **aligned** sub-GCPs perform best. Performance drops with simplified (-7.26%) or misaligned (-8.67%) configurations. We provided 3 practical guidelines: share core patterns, use realistic sizes, and include intermediate complexity.

**6. GNN Static Training vs. RL Dynamic Inference (R2)**
**Concern:** Discrepancy between fixed training structures and dynamic RL exploration.

**Response:** We clarified (Sec 3.2.3) that Forward Diffusion trains GNNs on **progressively noised graphs**. This noise injection acts as rigorous data augmentation, ensuring the GNN learns robust features that generalize well to the "imperfect" structures generated by RL actions during inference.

**7. Code Availability (R2)**
We are fully committed to reproducibility. We guarantee that the complete source code, data generation scripts, and pretrained models will be made publicly available via a GitHub repository upon acceptance.

**8. Instance Selection (R3)**
We clarified (Sec 4.1) that NSP instances 10-21 were selected to represent the "medium-to-large" complexity range (20-120 employees), following standard baseline protocols (e.g., MEO-HFG).

**Summary of Key Revisions**
*   **New Tasks:** FJSP (80 instances), 2E/3E-LRP (24 instances).
*   **New Analyses:** Feasibility (Table 4), Efficiency (Fig 8), Sub-GCP Sensitivity (Table 5).
*   **New Details:** Notation table, Action space specs, Weight evolution plot.

We believe these revisions comprehensively address the reviewers' concerns.

---

### Meta-Review · Area_Chair_5jxY · 2026-01-04

**Summary:**

The authors proposed a framework that combines graph diffusion models with reinforcement learning to address multilevel graph combinatorial problems.

The reviewers generally found the proposed idea novel and compelling, and the reported results are promising.

However, the overall support for acceptance is insufficient due to the following concerns:
- W1. Most importantly, the paper requires substantial improvement in presentation. Key algorithmic details are difficult to follow and, in some cases, underspecified.
- W2. The method does not provide any feasibility guarantee.
- W3. The framework incurs high computational cost, raising scalability concerns.
- W4. The experimental evaluation is limited to few settings, and the results are often difficult to interpret.
- W5. The submission is not accompanied by code.

Overall, while the proposed direction is promising, the current version of the paper is premature for acceptance at ICLR.

**Reviewer Concerns:**

The authors made substantial improvements addressing W2, and partial improvements for W3 and W4. However, significant room for improvement remains with respect to W1, and W5 remains unaddressed.

**Reviewer Scores:**

Even if some reviewers may slightly revise their evaluations upward, the paper is unlikely to receive enough overall support for acceptance.

---

### Decision · Program_Chairs · 2026-01-26

Reject